# Small-world complex network generation on a digital quantum processor

Eric B. Jones [1,2] ✉, Logan E. Hillberry [3], Matthew T. Jones[4,5], Mina Fasihi[4], Pedram Roushan[6], Zhang Jiang[6], Alan Ho[6], Charles Neill [6], Eric Ostby[6], Peter Graf[1], Eliot Kapit[4,7] ✉ & Lincoln D. Carr [4,7] ✉

Quantum cellular automata (QCA) evolve qubits in a quantum circuit depending only on the states of their neighborhoods and model how rich physical complexity can emerge from a simple set of underlying dynamical rules. The inability of classical computers to simulate large quantum systems hinders the elucidation of quantum cellular automata, but quantum computers offer an ideal simulation platform. Here, we experimentally realize QCA on a digital quantum processor, simulating a one-dimensional Goldilocks rule on chains of up to 23 superconducting qubits. We calculate calibrated and error-mitigated population dynamics and complex network measures, which indicate the formation of small-world mutual information networks. These networks decohere at fixed circuit depth independent of system size, the largest of which corresponding to 1,056 two-qubit gates. Such computations may enable the employment of QCA in applications like the simulation of strongly-correlated matter or beyond-classical computational demonstrations.

One of the most profound observations regarding the natural world is that, despite the simple set of physical laws that underpin it, the universe displays a plethora of complex, emergent phenomena, encountered in fields as diverse as biology, sociology, and physics[1–3]. Examples of classical systems where complexity arises as a result of many interacting degrees of freedom are ecosystems, the human brain, and power grids[4]. Certain classical cellular automata (CA) show how complexity can arise from simple rules without the controlling hand of a designer[5]. CA possess the ability to generate oscillatory, self-replicating structures and in some instances are themselves Turing complete[6–10].

It is known, however, that the laws constituting our best model of the universe are quantum mechanical rather than classical[11]. Therefore, in order to simulate the emergence of complexity more fundamentally, one ought to investigate computational models that are predicated upon quantum mechanics. Goldilocks quantum cellular automata (QCA)[12], are a class of computational models that exhibit emergent complexity despite being constructed from repeated blocks of simple local unitary operators[13]. They involve trade-offs in the local

neighborhood such as are known to be sources of complexity in classical systems and essential to self-organized criticality[14]. Some Goldilocks QCA have been shown to generate mutual information networks that exhibit signatures of complexity, such as large network clustering, short average path length, and broad node-strength distribution, typically only observed in classical, small-world networks like social or biological networks[12]. In addition, QCA have been proposed for other applications such as lattice discretization in the simulation of strongly-correlated matter, quantum field, and gravitational theories[15–17]. However, the categorical limitation on the ability of classical computers to simulate the time evolution of large quantum systems is a bottleneck for the discovery and exploration of QCA more generally, hampering the theoretical illumination of the class of systems as a whole[18]. Meanwhile, the last few years have seen the creation of sizeable digital quantum processors that are already demonstrating their value as tools of scientific discovery[19–25]. Due to their universality, such processors are ideal platforms on which to elucidate the physics and complexity characteristics of QCA.

[1]National Renewable Energy Laboratory, Golden, CO 80401, USA. [2]ColdQuanta Inc., Boulder, CO 80301, USA. [3]Department of Physics, University of Texas, Austin, TX 78712, USA. [4]Department of Physics, Colorado School of Mines, Golden, CO 80401, USA. [5]NVIDIA Corporation, Boulder, CO 80302, USA. [6]Google Quantum AI, Santa Barbara, CA 93117, USA. [7]Quantum Engineering Program, Colorado School of Mines, Golden, CO 80401, USA. ✉ e-mail: eric.jones@coldquanta.com; ekapit@mines.edu; lcarr@mines.edu

Herein, we simulate a particular one-dimensional QCA on a Sycamore-class superconducting processor depicted schematically in Fig. 1a–d. Through the calculation of population dynamics and a complex network characterization of the two-body mutual information matrix, we establish that such QCA form small-world mutual information networks and thereby exhibit emergent physical complexity. Our results contribute to enabling the widespread use of near-term quantum processors as QCA simulators and offer a template for how to experimentally investigate QCA generally. We note that complex network analysis has already made a largely, though not wholly, theoretical impact on quantum information. One example arises in one-way quantum computing[26] in which complex, network-structured graph states[27] are irreversibly transformed using projective measurements. Another occurs in the context of the quantum internet where communication channels between geographically-distant quantum devices are either imposed[28] by fiber optic networks or implied[29] by satellite downlink capabilities. Notably, satellite-based quantum communication channels have recently been shown to support small-world connectivity[30]. Our work differs significantly from these examples. Where one-way quantum computing has experimentally realized complex graph states by design via projective measurement, our work shows that QCA dynamically generate them in an emergent fashion. Where the quantum internet considers geographic networks, our networks of correlations emerge from unitary dynamics without any notion of physical distance except the locality of interactions. Finally, the complex networks occurring herein emerge on a generally-programmable, gate-model quantum processor, that is, an experimental platform with real-world constraints, such as processor noise, not necessarily present in prior theoretical work and different in nature than in prior experimental work.

## Results

### Quantum cellular automata

A one-dimensional (1D) quantum elementary cellular automaton may be defined as a chain of $L$ quantum bits (qubits) whose states are updated according to repeated blocks of neighborhood-local unitary operations along a discrete time axis. When every qubit's state has been updated, a QCA cycle, $t$, is completed. After selecting 1D chains of high-quality qubits from the available hardware graph (Fig. 1a; see also Supplementary Note 2), the structure (Fig. 1b) of a 1D QCA experiment is comprised of an initialization step, followed by the application of some number of QCA cycle unitaries out to cycle $t \in \{0, 1, \ldots, t_{max}\}$, and the measurement of appropriate observables.

The particular QCA that we simulate is the totalistic, three-site Goldilocks rule $T_6$ with a uniform Hadamard activation unitary applied to each qubit and boundary conditions equivalent to fixed $|0\rangle$s (see Supplementary Note 1)[12]. We note that the QCA notation $T_6$ should not be confused with decoherence times, which we will denote $\tilde{T}_1$ and $\tilde{T}_2$ where applicable. Figure 1c shows how rule $T_6$ is compiled down to quantum gates. A single, central bit flip initialization is followed by one QCA cycle, and finally, measurement in the computational $z$-basis. The local update, represented by two non-Clifford CH gates (green box), does nothing if there are zero or two adjacent $|1\rangle$s and applies the Hadamard activator to the central qubit if there is exactly one adjacent $|1\rangle$: this is the trade-off rule that gives rise to the Goldilocks nomenclature.

### Population dynamics and error mitigation

The quantum processor on which we run our QCA simulations is a 53-qubit superconducting processor, Weber, which follows the design of the Sycamore architecture outlined in ref. 20 (see also Supplementary Note 2). Typical performance characteristics for Weber are: single-qubit gate error $e_1 \approx 0.1\%$, two-qubit gate error $e_2 \approx 1.4\%$, $|0\rangle$-state readout error $e_{r0} \approx 2\%$, $|1\rangle$-state readout error $e_{r1} \approx 7\%$, and population relaxation time $\tilde{T}_1 \approx 15\,\mu s$[31]. Fig. 1d shows the decomposition of a single QCA cycle (red box) to the native $\sqrt{\text{iSWAP}}^\dagger$ two-qubit and $\text{PhXZ}(a, x, z) \equiv Z^z Z^a X^x Z^{-a}$ family of single-qubit gates. Strictly speaking, the native two-qubit gate is better modeled by $\sqrt{\text{iSWAP}}^\dagger \times \text{CPHASE}(\varphi)$, where the parasitic cphase is $\varphi \approx \pi/23$[21]. We

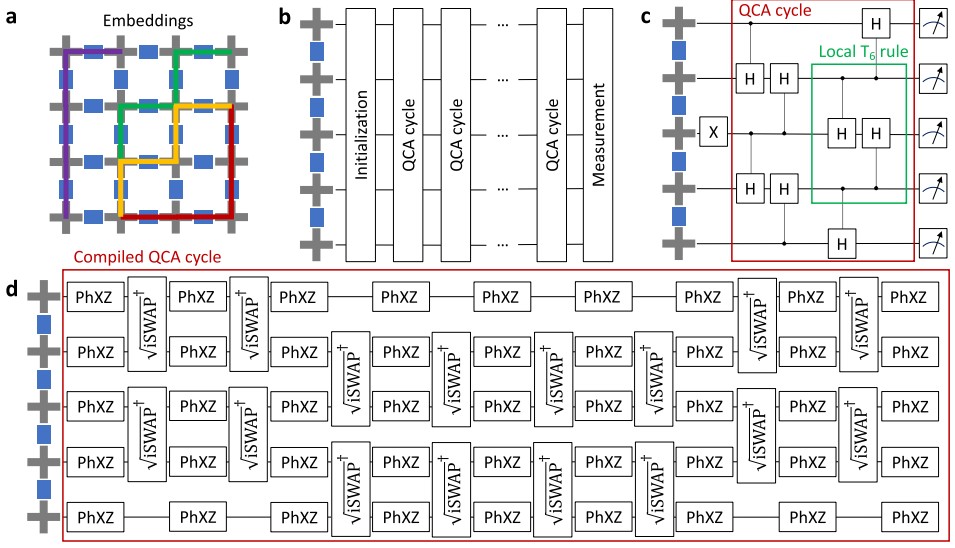

**Fig. 1 | One-dimensional quantum cellular automata circuits. a** Schematic for embedding one-dimensional chains into a subset of a two-dimensional Sycamore-class quantum processor. Gray crosses represent transmon qubits and blue rectangles represent couplers. Purple, green, yellow, and red paths are hypothetical example embeddings. **b** Generic structure of a one-dimensional quantum cellular automata (QCA) circuit where time flows to the right. An initialization step is applied to a chain of $L$ qubits, typically to place them into a classical product state with some number of bit flips ($|1\rangle$s). A number of unitary QCA update cycles, $t$, are applied to all $L$ qubits before a measurement is performed. **c** The specific structure of a Goldilocks QCA for one QCA cycle (red box), wherein the initial state is $|0\ldots010\ldots0\rangle$, the local update unitary is a controlled Hadamard gate, and measurement is performed in the computational basis. **d** After moment alignment, spin-echo insertion, and compilation down to hardware-native gates a single QCA cycle (red box) results in $4 \times (L-1)$ number of $\sqrt{\text{iSWAP}}^\dagger$ gates and $8 \times L$ number of individually-parameterized $\text{PhXZ}(a, x, z) \equiv Z^z Z^a X^x Z^{-a}$ gates. The number of single and two-qubit layers per QCA cycle does not change as a function of system size, only total gate volume does.

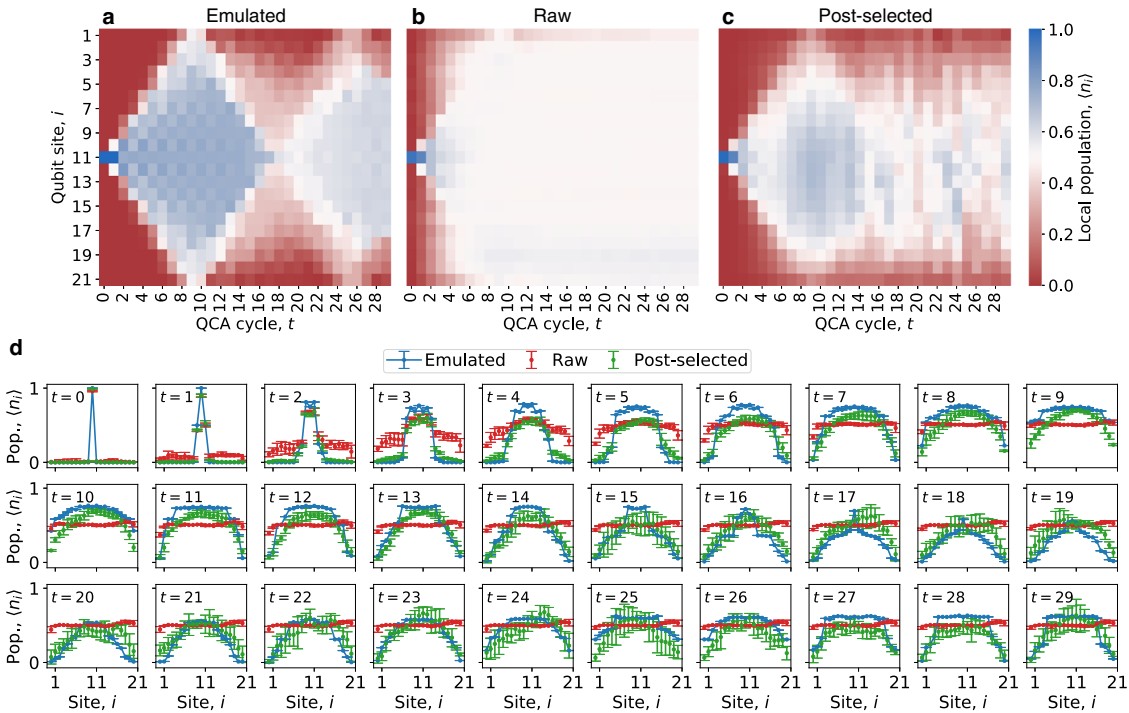

**Fig. 2 | Population dynamics and post-selection.** A noise-free, numerically-emulated Goldilocks QCA out to 30 cycles, initialized with a single $|1\rangle$ in the middle of the 21-qubit chain. Coherent local population, $\langle n_i \rangle$, dynamics that resist equilibration can be observed in the form of the two blue diamond shapes. **b** Raw population dynamics for the same QCA circuit averaged over four 21-qubit chains embedded into the 53-qubit Weber processor. **c** The same experimental data as **b** but with post-selection applied based on domain wall conservation. **d** Time-stamped population vignettes show the same dynamics quantitatively for emulated (blue- lines for visual clarity), raw (red), and post-selected (green) data. Error bars represent one standard deviation from the mean over four different chains.

apply a suite of low-overhead circuit optimization, calibration, and error mitigation techniques to optimize circuit performance including moment alignment, spin-echo insertion, Floquet calibration[21,24], parasitic cphase compensation, and most importantly, post-selection (see Supplementary Note 4).

At each QCA cycle depth we measure the output of the circuit in the $z$-basis $N_c = 100,000$ times, resulting in a set of $L$-bit strings $\{|z\rangle\}$ and associated probabilities $\{P_z \approx N_z/N_c\}$, where $N_z$ is the number of times bit string $|z\rangle$ is observed. The local population on each site is calculated via $\langle n_i \rangle = (1 - \sum_z P_z(-1)^{z_i})/2$ and averaged over four 1D qubit chains. Figure 2a shows the numerical emulation of such a procedure initialized with a single, central bit flip on 21 qubits out to 30 QCA cycles. The two large-scale blue diamonds indicate coherent dynamics. When repeated on the Weber processor, a combination of photon loss, gate error, and state preparation and measurement (SPAM) error leads to nearly total population decoherence by $t \approx 10$ (Fig. 2b). We therefore post-select experimental data and discard any measurements whose eigenvalues of the Ising-like operator $\mathcal{O} = \sum_{i=0}^{L} Z_i Z_{i+1}$ differ from the corresponding eigenvalue of the initial state. That is, $\mathcal{O}$ is a dynamical invariant of the $T_6$ rule that keeps track of the number of domain walls in the system. Figure 2c shows that post-selection results in coherent population dynamics that persist beyond $t \approx 10$, although different observables can degrade with noise on slightly different timescales (see Fig. 3). The cycle-stamped population vignettes shown in Fig. 2d support these observations more quantitatively, with error bars representing one standard deviation on the four different chains. After $t \approx 15$, error bars on the post-selected data become more significant and while some qualitative features of the emulated population dynamics appear to persist, such as a larger population towards the center of the chain, it is unclear from Fig. 2 alone as to what the underlying nature of these qualitative similarities is. Moreover, our complex network analysis of the behavior of rule $T_6$ relies on the calculation of two-body observables beyond the one-body observables

depicted in Fig. 2. As such, we turn to a calculation of Shannon mutual information both to more deeply understand the long-time population dynamics of our QCA and to establish their complex network behavior.

## Mutual information network analysis

Following the complex-network approach in neuroscience wherein functional connectivity of the brain is characterized between spatially non-adjacent regions[32], we calculate the classical, Shannon mutual information between all pairs of qubits in each 1D Goldilocks QCA chain

$$I_{ij} \equiv \sum_{z_i=0}^{1} \sum_{z_j=0}^{1} p(z_i, z_j) \log_2 \frac{p(z_i, z_j)}{p(z_i)p(z_j)} \qquad (1)$$

and regard it as an adjacency matrix of correlations that defines the QCA network at each cycle. We choose to use classical, rather than a measure of quantum, mutual information because its calculation only requires measurements in the computational $z$-basis, which we have shown are amenable to post-selection. Moreover, we show in Supplementary Note 5 that the Shannon (classical) mutual information acts as a reliable proxy for von Neumann (quantum) mutual information for the $T_6$ QCA.

Complex networks are ones that are neither purely regular, such as a lattice or complete graph, nor entirely random[33]. The classic demonstration that a network has complex, small-world character involves showing persistently large clustering and simultaneously short path length[34], with a power-law node strength distribution resulting in highly-connected nodes. By analogy with transportation networks, these features describe networks that are easily traversed both locally (clustering) and globally (path length), and exhibit hubs (broad node strength distribution). While transportation networks provide an intuitive interpretation of these complex network measures, we emphasize that clustering, path length, and node strength

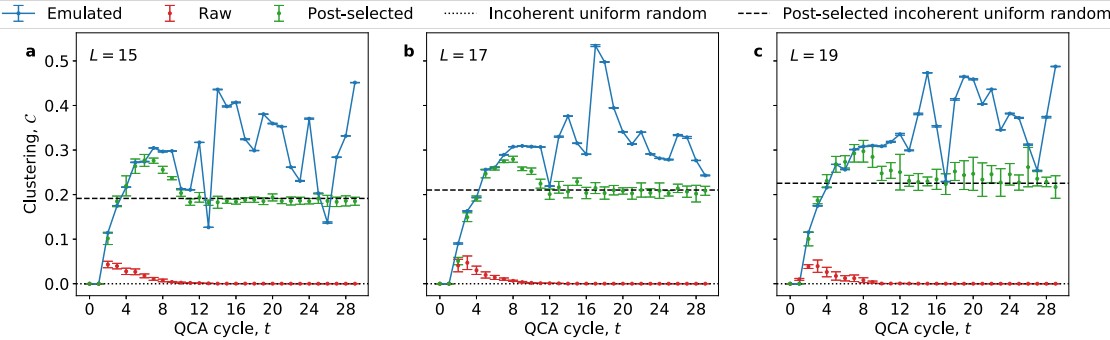

**Fig. 3 | QCA mutual information network clustering.** Clustering coefficient as a function of QCA cycle for three intermediate system sizes simulated on Weber, $L = 15$ (**a**), 17 (**b**), and 19 (**c**). Blue curves (lines for visual clarity) are calculated from numerical emulation, while red (green) data points are from Weber data without (with) post-selection. Error bars are one standard deviation in $\mathcal{C}$ over four different chains. Dashed (dotted) black lines are the clustering of the incoherent uniform random state with (without) post-selection.

distribution have seen widespread use in analyzing the structure of mutual information networks and in drawing conclusions regarding the physical complexity of the underlying system in both classical and quantum systems. For example, applying these measures to mutual information networks derived from electroencephalographic or fMRI data has been used to elucidate structure-function correlations in the brain[32,35]. In addition, the measures have been used along with earthquake time series-derived mutual information to model seismicity[36] and with mutual information in wireless networks to explain routing efficiency[37]. Finally, complex network measures calculated on quantum mutual information networks in quantum Ising and Bose–Hubbard models are able to detect quantum phase transition critical points[38]. Hence, the use of clustering, path length, and node strength distribution in conjunction with mutual information is a well-established, quantitative procedure with predictive power. We employ this procedure in order to understand the structure of correlations in our QCA circuits and to observe the emergence of physical complexity in the presence of quantum processor noise.

Through decoherence, the state of a quantum processor approaches an incoherent uniformly random state with all amplitudes equal to $2^{-L/2}$ and for which $I_{ij} = 0$ for all $L \geq 2$. Hence, the incoherent uniformly random state is neither locally nor globally traversable and is thus not a typical random network. It is also useful to consider the effect of subjecting the incoherent uniformly random state to the same post-selection procedure as our experimental data. This can be done in three steps: (i) form an incoherent uniformly random state with all $2^L$ amplitudes equal to $2^{-L/2}$, (ii) set any basis state amplitudes to zero if the basis state has an eigenvalue under $\mathcal{O}$ that differs from that of the QCA's initial condition, and (iii) renormalize the remaining amplitudes so that the state vector has unit norm. Upon post-selection, this decohered state is no longer uniform and the corresponding mutual information network is both non-zero and non-random, although to a much lesser extent than the states generated by Goldilocks QCA. The complexity of networks generated by Goldilocks QCA is established by computing network measures for emulated, raw processor, and post-selected processor states and then comparing each of these to post-selected incoherent uniform random states.

Clustering measures local network transitivity and is defined as the ratio of the weighted number of closed triangles in the network to the weighted total number of length-2 paths in the network (i.e., the number of potentially closed triangles—see Supplementary Note 6). The first relevant signature of network complexity is intermediate to large clustering values that do not decay with system size, in contrast to random networks. The emulated clustering (blue curves) of the QCA exhibits this signature and actually increases slightly as a function of system size, $L$ (see Fig. 3). While we plot three of the larger system sizes

we simulated here, $L = 15$, 17, and 19, this proves true for all other system sizes simulated as well. Next, we note that without post-selection, the clustering $\mathcal{C}$ calculated from raw data from Weber (red points) rises briefly but then quickly decays toward zero, the incoherent uniformly random limit (black dotted curve), at $t \approx 12$ for all three system sizes. In contrast, the green curves in Fig. 3 show that with post-selection the experimental clustering tracks the emulated clustering closely until $t \approx 6$ and remains larger than post-selected uniform randomness (black dashed curve) until $t \approx 12$, independent of system size. There is therefore a window between $t \approx 4$ and 12 over which we can analyze the formation of a non-random complex network in the QCA for all system sizes simulated. We provide a more detailed description of our cycle windowing process in Supplementary Note 7.

Figure 4a shows the coherence window, cycle-and-chain-averaged emulated (blue), raw (red), and post-selected (green) clustering coefficient for $L = 5$–23 qubits. After the finite-size effects encountered for $L \leq 11$, it is clear that while the raw clustering trends towards zero—that of an incoherent uniformly random state network —both the emulated and post-selected clustering stabilize towards $\mathcal{C} \approx 0.3$ and trend towards larger values as a function of system size, indicating substantial network transitivity beyond post-selected randomness (black dashed curve). Figure 4b shows the coherence window, cycle-and-chain-averaged weighted shortest path length, $\ell$, as a function of system size, which gauges global network traversability (see Supplementary Notes 6 and 7). The raw data path length (red) in Fig. 4b is large and increases as a function of system size. The post-selected (green) path length tracks the emulated (blue) path length closely, trends downward, and is always one to two orders of magnitude smaller than the raw path length. Interestingly, post-selected path length tracks the path length of post-selected randomness (black dashed line) nearly as well as it does emulated path length. Taken together, however, Fig. 4a, b signal the existence of small-world mutual information networks generated in the coherence window of the Goldilocks QCA beyond what can be obtained by post-selecting incoherent uniform randomness. The end of the coherence window ($t = 12$) for the largest system size simulated ($L = 23$) corresponds to 1056 $\sqrt{\text{iSWAP}}^{\dagger}$ gates.

Figure 4c further indicates the formation of small-world mutual information networks, showing that the size-normalized emulated (blue) and post-selected (green) node strength distributions (see Supplementary Note 6), $P[g_i/(L-1)]$, are relatively flat between $1 \times 10^{-2}$ and $2 \times 10^{-1}$ compared with those of the post-selected random node strengths (black dashes), which peak between $-2.5 \times 10^{-2}$ and $1.5 \times 10^{-1}$, and raw (red) node strengths, which are heavily biased towards much smaller values, indicating a deficit in network connectivity. Finally, Fig. 4d–g visually depicts how the mutual information for $L = 23$ at $t = 9$ differs for the raw QCA data, which approaches the network structure

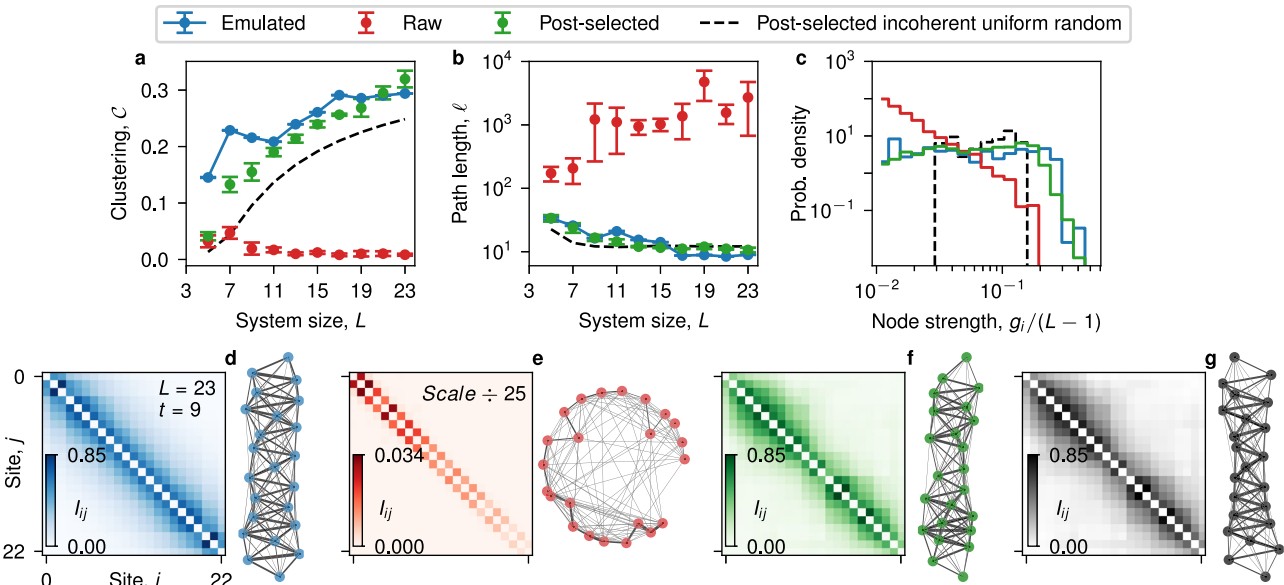

**Fig. 4 | Small-world mutual information network behavior. a** Coherence window, cycle-and-chain-averaged clustering, $\mathcal{C}$, as a function of system size, $L$. **b** Coherence window, cycle-and-chain-averaged path length, $\ell$, as a function of system size, $L$. **c** Normalized node strength distribution, $P[g_i/(L-1)]$, amalgamated over all system sizes. **d**–**g** Heatmap and force-directed[41] complex network visualization of mutual information network for $L = 23$ at $t = 9$ (blue: emulated; red: raw data from Weber; green: post-selected; black: post-selected incoherent uniform randomness). Where present, error bars represent one standard deviation from the mean averaged over four different chains and cycles within the coherence window.

of incoherent uniform randomness, and the emulated and post-selected QCA networks, which both display lattice girder-like small-world structure that resemble one another more closely than they do post-selected randomness.

**Towards beyond-classical QCA**

In addition to their intrinsic scientific value as quantum models for emergent complexity, QCA also present intriguing prospects for establishing new inroads into the beyond-classical era. In the instance of Goldilocks rule $T_6$, identification of a dynamical invariant makes simulation less fragile to noise than a fully chaotic random quantum circuit (RQC). However, it also implies efficient—that is, polynomial—classical emulation via direct Schrödinger evolution since it reduces the bounded size of the protected $T_6$ Hilbert space to scale as ~$0.63 L^{1.91}$ for a fixed, single initial bit flip at the center of the chain (see Supplementary Note 4). Generally, fixing the number of initial $|1\rangle$s while increasing system size leads to polynomial space and time complexity. However, one can recover exponential scaling in the protected Hilbert space by simulating increasingly large chains while fixing the density (rather than number) of initial $|1\rangle$s well separated by $|0\rangle$s. For a fixed density, $\rho_{|1\rangle s}$, the protected Hilbert space bound scales as the binomial coefficient, ~$\binom{L}{\rho_{|1\rangle s} L}$, which asymptotically scales exponentially in $L$. For context, the long-time, cycle-averaged bond entropy of rule $T_6$ was shown to scale between a 1D area and volume law[12]. Although simulation of Goldilocks rules has shown the failure of direct matrix-product-state approaches[39], given this intermediate scaling it is an open problem as to whether efficient classical simulatability may be achieved using a modified tensor network approach in the absence of a dynamical invariant[12,40]. Moreover, efficient simulatability of area law-scaling states in two-dimensions (2D) or higher using tensor network approaches, while promising, is even less assured than in 1D. Hence, 2D QCA that exhibit area-law scaling or worse may be good candidates for beyond-classical demonstrations.

Here we have demonstrated that existing quantum processors can simulate 1D QCA with high fidelity at large gate volume. While reliant on the availability of high-fidelity hardware, the main circuit design principles that enable this goal consist of the identification of particular QCA rules that: (i) generate significant complexity

signatures, (ii) efficiently compile to low-depth sequences of hardware-native gates, and (iii) are amenable to post-selection through identification of one or more dynamical invariants (For a more complete discussion of the quantum-simulational complexity of the $T_6$ rule including post-selection overhead, please see Supplementary Note 4). We therefore expect these design principles to aid in discovering QCA that support beyond-classical demonstrations or are otherwise useful in quantum computational domain applications. In particular, employing such design principles for QCAs that model correlated quantum matter could be a promising route toward beyond-classical simulation of novel physical systems in the near term.

## Data availability

The data supporting this work are available at the public Dryad repository (https://doi.org/10.5061/dryad.fbg79cnxd).

## Code availability

The code supporting this work is available at the public GitHub repository (https://github.com/ebjones424/qca_nat_comms).

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

## Acknowledgements
We thank the Google Quantum AI team. This work was supported in part by the NSF under grants PHY-1653820 (to E.K.) and DGE-2125899, CCF-1839232, PHY-1806372, and OAC-1740130 (to M.F., M.T.J., L.D.C.). This work was authored in part by the National Renewable Energy Laboratory (NREL), operated by Alliance for Sustainable Energy, LLC, for the U.S. Department of Energy (DOE) under Contract No. DE-AC36-08GO28308. This work was supported in part by the Laboratory Directed Research and Development (LDRD) Program at NREL (to E.B.J., P.G.). The views expressed in the article do not necessarily represent the views of the DOE or the U.S. Government.

## Author contributions
Conceptualization: E.K., L.D.C. Data curation: E.B.J., L.E.H., M.T.J., M.F., Z.J., A.H. Formal analysis: E.B.J., L.E.H., M.T.J., M.F., P.R., Z.J., A.H., C.N., E.O., E.K., L.D.C. Funding acquisition: E.B.J., P.G., E.K., L.D.C. Investigation: E.B.J., L.E.H., M.T.J., M.F., P.R., Z.J., A.H., C.N., E.O., E.K., L.D.C. Methodology: E.B.J., L.E.H., M.T.J., M.F., P.R., Z.J., A.H., C.N., E.O., E.K., L.D.C. Project administration: P.R., A.H., E.O., E.K., L.D.C. Resources: P.R., Z.J., A.H., C.N., E.O. Software: E.B.J., L.E.H., M.T.J., Z.J., A.H. Supervision: P.G., E.K., L.D.C. Validation: all authors. Visualization: E.B.J., L.E.H., P.R., Z.J., A.H., C.N., E.K., L.D.C. Writing—original draft: E.B.J. Writing—review and editing: all authors.

## Competing interests
The authors declare no competing interests.
