## [Peer Review File · Nature Communications]

Small-world complex network generation on a digital quantum processorREVIEWER COMMENTS

Reviewer #1 (Remarks to the Author):

The authors demonstrated that a Goldilocks QCA can be efficiently simulated on a medium-scale quantum computer. They have been very cautious and analyzed all possible limits of their results by statistical classical means. They also remarked a complexity signature that has never been noticed in the theoretical framework. Their simulations are convincing and the supplementary material robustly support the reader through their calculations. I really enjoyed read this manuscript and, as a researcher actively involved in this topic, I strongly believe that such results will be largely considered by the QCA community. I believe they authors provided successfully the evidence that a non trivial QCA (Goldilocks QCA are known to be very reach, although 'simple') can be efficiently simulated by a NISQ computer.

Said that, I strongly recommend the publication of this paper as it is.

Reviewer #2 (Remarks to the Author):

The submitted manuscript reports on complex networks of correlations between qubit pairs on a superconducting processor of the Sycamore class. The framework are quantum cellular automata (QCA) as the quantum circuit does produce the dynamics of a QCA reported by the authors in Ref. [22], named T6 or Goldilocks. The theoretically predicted unitary dynamics of the circuit lasts shortly on the experimental platform and after few cycles a maximally mixed state is reached. Through post-selection it is possible to extract a coherent dynamics up to a longer time and to observe the build-up of correlations before coherence fades. The main focus of the manuscript is on the observation in a superconducting computer (with post-selection) of a small-world complex network of classical correlations. In the following I address the main results in more detail and their novelty/strength.

1) Classical and quantum correlation networks

The possibility to generate complex mutual information networks is well established in classical systems and also in quantum ones. In particular here the authors consider correlations as encoded in the Shannon mutual information and show (in the Suppl. Information) deviations between classical and quantum mutual information matrices of about 10%. Indeed what is experimentally measured is the network of 'classical' correlations between quantum bit pairs -- because post-selection does not allow to access quantum mutual information.

Looking at the state of the art, the authors could mention complex networks of 'quantum' correlations that have been considered in other theoretical works (including and not limited to Ref.[22]). Also relevant examples have been reported in the context of one way quantum computing (see cluster states broad literature) and in the quantum internet, where also small world networks have been specifically considered (see e.g. the recent arXiv:2012.01125, PRX Quantum).

2) Complex and random networks.

In the present approach correlations not present in the initial state do emerge during the evolution. This is of course a typical scenario reported in several dynamical systems ranging from atomic to photonic ones. As a peculiarity, here the self-repeating circuit detailed in Fig.1 produces the dynamics of the T6 QCA. Cellular automata are characterized by rather complex dynamics even if governed by simple evolution rules and quantum CA have been an active field from the 80's. In the T6 QCA the expected dynamics (in absence of decoherence) can be visualized a slow decaying simple spatio-temporal rhomboid pattern, a shape determined by the presence of boundaries (finite spin chain).

Two-qubit gates do typically induce some correlations and produce different complex networks in circuits. In particular the T6 and other QCA have already been analysed in Ref. [22]. This raises some concern about the novelty and strength of the message. In most of the situations

correlations networks are neither simply random (due to the physical dynamics creating some "structure") nor perfectly regular (unless restricted to highly symmetric systems and states). Therefore the emergence of complex networks is not a breakthrough by itself.

3) Experimental classical correlation network.

The experimental implementation in the Sycamore platform is the major novelty of this work. Chains between few and more than 20 qubits are considered.

Looking at the dynamics of clustering and path length for the classical mutual information matrix (Fig.3), a temporal window (between 4 and 12 cycles) is observed in which both quantifiers agree in the experimental post-selected and in the theoretical prediction, while in absence of post-selection the state becomes fully mixed. Post selection is indeed needed to get a mutual information network. The price to pay is the reduced number of selected runs out of the total, a very short time window of agreement before reaching a state completely determined by post selection, and access to only classical mutual information. Actually post-selection by itself is apparently enough to induce complex networks even starting from uniform random states. This manifests in the growing clustering (dashed line in Fig.4 a), rather significant for these states even if smaller than for the T6 post-selected ones.

It would also be worth to comment on the platform choice, given the limitations imposed by decoherence, spoiling any correlations in few cycles and allowing to observe only classical correlations through post-selection. The authors could comment on the choice of this platform and on which performance could be expected in an atomic one, as envisaged in other works (e.g. Rydberg atoms).

In conclusion, the manuscript is overall well presented and sound, the topic and the experimental realization are interesting and original and it deserves to be published. On the other hand, Nature Communication is rather selective.

The main novelty of this work is the experimental detection of a classical network of correlations in post-selected states of a QCA. Post selection represents a strong limitation in the sense that actually even post-selected incoherent uniform random states would lead to a clustering of the order of (even if smaller than) the experimentally reported one.

I am therefore recommending this paper for a more specialized journal

Reviewer #3 (Remarks to the Author):

This paper describes an experimental study of quantum cellular automata (QCA) using superconducting qubits. QCA can produce complex quantum states using only geometrically-local operations applied repeatedly. The authors realize a particular QCA comprising controlled-Hadamard gates, and show that the observed evolution in the computational basis agrees quite well with noiseless simulations. This agreement depends critically on post-selection, where measurement results that do not respect a known symmetry of the QCA, due to errors, are discarded. The authors then measure 2-site correlations in the computational basis between all pairs of qubits, and construct matrices of the resulting Shannon mutual information. Interpreting these as graph adjacency matrices, they analyze the correlations in terms of various quantities which arise in the study of complex networks. From these they conclude that the QCA is indeed creating complex quantum states from simple operations.

I find this paper well-written and quite elegant. As far as I can tell, the experimental methods and the analyses are generally sound. More broadly, the motivation for studying QCA is explained in a compelling way, and the experimental results look quite good. Unfortunately, I am not qualified to comment on the relation of this paper to previous work on QCA or network analysis, nor on the gate calibrations or prospects for classical simulation mentioned at the end. I trust that the other

referees will be able to vet these aspects in more detail.

While I think this is nice paper, I am on the fence about whether it should be published in this journal or in a more specialized one. I have a few major concerns, as well as some more minor ones.

Major concerns:

- The authors claim that this is the first experimental realization of QCA on a digital quantum processor. However, any repeated application of identical 1- and 2-qubit gates seems to fit the broad definition of QCA given in the main text. For instance, a Trotterized simulation of a 2-local Hamiltonian (e.g., of a transverse-field Ising model or a Heisenberg model in 1D) seems very similar to, if not equivalent to, a QCA. Trotter circuits have certainly been realized before, and also generate complex quantum states using repeated local operations. There is nothing exotic about that, as far as I can tell. Besides the interpretation, where exactly lies the boundary between this experiment and earlier ones with a similar structure?
- It seems reasonable enough to interpret 2-qubit correlations as describing a graph. However, there's a conceptual leap made in the second half of the paper wherein the system starts to get treated like an actual network on this basis. The authors rely heavily on this analogy, and use it to motivate several figures of merit, including clustering, shortest path length, and node strength distribution. In the supplemental material these quantities are explicitly explained in terms of transportation networks. But the underlying quantity in these experiments is not a transportation network or the like; it is simply 2-point correlations between a set of binary random variables. It is not obvious to me that these network-inspired figures of merit are appropriate/meaningful for analyzing such correlations, and for concluding that the underlying states are complex. In fact, the matrices in Figs. 4d-g don't look particularly exotic until they're turned into seemingly-abstruse figures of merit. Have these network-inspired quantities been used to study correlations in other papers (quantum or otherwise)? The authors should explain in more detail why a reader should care about the quantities plotted in Figs. 3 and 4.
- On a similar note, I'm uneasy about the use of classical, as opposed to quantum, mutual information throughout. I understand that it would probably not be feasible to measure the quantum mutual information, and I am not suggesting that this is essential. But I'm worried that the classical mutual information could depend heavily on the measurement basis. It would be reassuring to see, even if just in a small simulation (no need for further experiments), that the same conclusions follow from local measurements along the x- or y-axes, not just the z-axis.

Minor concerns:

- The authors claim in their conclusion that existing quantum processors can efficiently simulate 1D QCA. Yet, in the supplemental material (SM) they say that the retained count fraction (after post-selection) decays roughly exponentially with the number of qubits for fixed t . In other words, the time required to get a constant number of counts increases exponentially with the system size. How should I reconcile these two statements? Also, it would be helpful to add a plot to the SM showing the same data as in Fig. (SM-7) but with L on the x axis for fixed values of t . (Unless I've missed it somewhere.)
- What's a "post-selected incoherent uniform random state"? Is it just post-selected counts from the maximally mixed state (i.e., from a uniform distribution)?
- Is there a reason for using the word "emulate" rather than "simulate" when the latter is more common?
- The authors claim that the system's dynamics occupy a fraction $\sim 1.08^L$ of the overall Hilbert space (due to the symmetry used for post-selection I believe). Is this just an upper-bound? Saying the state truly explores this region of the Hilbert space is a much stronger claim than simply saying it is constrained to this region.

- Finally, a couple of suggestions for the figures: It would be helpful to put "t=0" rather than just 0 (maybe likewise for the other numbers) in Fig. 2b. It's not clear at first glance what those numbers mean as-is. Also, it would be helpful to add labels to the color bars in Fig. 4d-g to spell out what quantity is being plotted.

Again, I think this is a nice paper, but I cannot recommend it for publication in this journal until some gaps are filled.

Reviewer Responses Round One

March 11, 2022

Reviewer #1 (Remarks to the Author):

The authors demonstrated that a Goldilocks QCA can be efficiently simulated on a medium-scale quantum computer. They have been very cautious and analyzed all possible limits of their results by statistical classical means. They also remarked a complexity signature that has never been noticed in the theoretical framework. Their simulations are convincing and the supplementary material robustly support the reader through their calculations. I really enjoyed read this manuscript and, as a researcher actively involved in this topic, I strongly believe that such results will be largely considered by the QCA community. I believe they authors provided successfully the evidence that a non trivial QCA (Goldilocks QCA are known to be very reach, although 'simple') can be efficiently simulated by a NISQ computer. Said that, I strongly recommend the publication of this paper as it is.

We thank the reviewer for a careful reading of our manuscript and their associated recommendation to publish.

Reviewer #2 (Remarks to the Author):

The submitted manuscript reports on complex networks of correlations between qubit pairs on a superconducting processor of the Sycamore class. The framework are quantum cellular automata (QCA) as the quantum circuit does produce the dynamics of a QCA reported by the authors in Ref. [22], named T6 or Goldilocks. The theoretically predicted unitary dynamics of the circuit lasts shortly on the experimental platform and after few cycles a maximally mixed state is reached. Through post-selection it is possible to extract a coherent dynamics up to a longer time and to observe the build-up of correlations before coherence fades. The main focus of the manuscript is on the observation in a superconducting computer (with post-selection) of a small-world complex network of classical correlations. In the following I address the main results in more detail and their novelty/strength.

We thank the reviewer for a careful reading of our manuscript and appreciate the opportunity to address their concerns and comments below.

- 1) Classical and quantum correlation networks. The possibility to generate complex mutual information networks is well established in classical systems and also in quantum ones. In particular here the authors consider correlations as encoded in the Shannon mutual information and show (in the Suppl. Information) deviations between classical and quantum mutual information matrices of about 10%. Indeed what is experimentally measured is the network of 'classical' correlations between quantum bit pairs – because post-selection does not allow to access quantum mutual information. Looking at the state of the art, the

authors could mention complex networks of ‘quantum’ correlations that have been considered in other theoretical works (including and not limited to Ref.[22]). Also relevant examples have been reported in the context of one way quantum computing (see cluster states broad literature) and in the quantum internet, where also small world networks have been specifically considered (see e.g. the recent arXiv:2012.01125, PRX Quantum).

We thank the reviewer for pointing our attention to instances of complex networks formed from quantum correlations in the *theoretical* literature. We emphasize that while our correlation networks are predicated on a classical measure of mutual information– due to a choice of measurement basis constrained by post-selection– our work breaks new ground in that it is the first instance of complex networks generated and measured directly on a generally-programmable quantum processor, that is, an *experimental* platform with real-world constraints, such as processor noise, not necessarily present in prior theoretical work. There is no substitute for a real experiment. We place the following in the introduction to our manuscript (highlighted in yellow):

“We note that complex-network analysis has already made a theoretical impact on quantum information. One example arises in one-way quantum computing [Phys. Rev. A 68, 022312] in which complex, network-structured graph states [Nature Physics volume 15, pages 148–153 (2019)] are irreversibly transformed using projective measurements. Another occurs in the context of the quantum internet where communication channels between geographically-distant quantum devices are either imposed [Phys. Rev. Lett. 124, 210501] by fiber optic networks or implied [C Bonato et al 2009 New J. Phys. 11 045017] by satellite downlink capabilities. Notably, satellite-based quantum communication channels have recently been shown to support small-world connectivity [PRX Quantum 2, 010304]. Our work differs significantly from these examples. Where one-way quantum computing assumes the existence of complex graph states, our work shows QCA generate them dynamically. Where the quantum internet considers geographic networks, our networks of correlations emerge from unitary dynamics without any notion of physical distance except the locality of interactions. Finally, the complex networks occurring herein emerge on a generally-programmable quantum processor, that is, an experimental platform with real-world constraints, such as processor noise, not necessarily present in prior theoretical work.”

- 2) Complex and random networks. In the present approach correlations not present in the initial state do emerge during the evolution. This is of course a typical scenario reported in several dynamical systems ranging from atomic to photonic ones. As a peculiarity, here the self-repeating circuit detailed in Fig.1 produces the dynamics of the T6 QCA. Cellular automata are characterized by rather complex dynamics even if governed by simple evolution rules and quantum CA have been an active field from the 80’s. In the T6 QCA the expected dynamics (in absence of decoherence) can be visualized a slow decaying simple spatio-temporal rhomboid pattern, a shape determined by the presence of boundaries (finite spin chain). Two-qubit gates do typically induce some correlations and produce different complex networks in circuits. In particular the T6 and other QCA have already been analysed in Ref. [22]. This raises some concern about the novelty and strength of the message. In most of the situations correlations networks are neither simply random (due to the physical dynamics creating some “structure”) nor perfectly regular (unless restricted to highly symmetric systems and states). Therefore the emergence of complex networks is not a breakthrough by itself.

Here we endeavor to address the reviewer’s concern in a point-by-point manner below.

i) **“In the present approach correlations not present in the initial state do emerge during the evolution. This is of course a typical scenario reported in several dynamical systems ranging from atomic to photonic ones.”** We agree that correlations of one type or another emerge during the dynamical evolution of interacting quantum mechanical systems generally. In the present work however, we focus on the emergence of correlations with a very specific, that is, small-world, structure, a structure that has heretofore not been demonstrated to survive for any length of time in the presence of digital quantum processor noise.

ii) **“As a peculiarity, here the self-repeating circuit detailed in Fig.1 produces the dynamics of the T6 QCA. Cellular automata are characterized by rather complex dynamics even if governed by simple evolution rules and quantum CA have been an active field from the 80’s. In the T6 QCA the expected dynamics (in absence of decoherence) can be visualized a slow decaying simple spatio-temporal rhomboid pattern, a shape determined by the presence of boundaries (finite spin chain).”** We clarify that while the particular pattern observed in the main manuscript for the T_6 dynamics is dictated by initial conditions and boundary conditions, *that* the T_6 QCA exhibits long-time, coherent dynamics that resist equilibration is a feature of the T_6 cycle update unitary and is robust under a variety of initial and boundary conditions [L. E. Hillberry, 2016, https://mountainscholar.org/bitstream/handle/11124/170336/Hillberry_mines_0052N_11082.pdf?sequence=1] [L. E. Hillberry et al., 2021, doi:<https://doi.org/10.1088/2058-9565/ac1c41>]. This is important, since by comparison, many other T_R rules with similar initial and boundary conditions display less persistent, or indeed very quickly-equilibrating, dynamics that relax to an incoherent state even without environmental noise. The use of the descriptor, “simple”, here is therefore subjective. While many one-dimensional classical cellular automata may appear more visually complex as gauged by their population dynamics, it is worth noting that the dynamics of the T_6 QCA, though oscillatory, have been shown numerically neither to be wholly repetitive nor fully chaotic, even at very deep QCA cycles [L. E. Hillberry, 2016, https://mountainscholar.org/bitstream/handle/11124/170336/Hillberry_mines_0052N_11082.pdf?sequence=1] [L. E. Hillberry et al., 2021, doi:<https://doi.org/10.1088/2058-9565/ac1c41>]. This observation is borne out in the dynamical complex network analysis of the T_6 mutual information network as well. This suggests that the closest analogous classification of T_6 classically would be the *class 4* cellular automata. From this perspective, the dynamics of rule T_6 are as “complex” as those observed in classical cellular automata. Given that the degrees of freedom in classical and quantum cellular automata are, by definition, very different in nature and that our quantum processor simulations of rule T_6 are constrained in size and accessible initial conditions by processor noise characteristics and the efficacy of post-selection, it is unrealistic to expect that the visually-assessed complexity of the dynamics of local observables (here population) should manifest in the same way for quantum cellular automata as it does for classical cellular automata.

iii) **“Two-qubit gates do typically induce some correlations and produce different complex networks in circuits. In particular the T6 and other QCA have already been analysed in Ref. [22]. This raises some concern about the novelty and strength of the message.”** We address this objection in two parts. First, while it is true that two-qubit gates typically do, and indeed are designed to, induce correlations between qubits, and while it *may* be true that two-qubit gates typically produce complex mutual information networks in quantum circuits, there is, to

our knowledge, no prior work showing such networks to have either been produced experimentally or characterized explicitly (via complex network analysis) on gate-model processors in the face of noise. Second, the reviewer raises a concern about the novelty of our work on the basis that the T_6 QCA was first predicted to demonstrate a complex mutual information network in some of our co-authors' prior theoretical work by simulating the T_6 dynamics *on a classical computer without noise* [L. E. Hillberry et al., 2021, doi:<https://doi.org/10.1088/2058-9565/ac1c41>]. Raising an objection to the novelty of our work based on such a concern is tantamount to a very general argument that just because a physical phenomenon is first predicted theoretically, there is little novelty in demonstrating the phenomenon experimentally. That argument, taken seriously, would disincentivize one of the primary avenues through which scientific knowledge is created: the experimental testing of a theoretical prediction or a theory-driven hypothesis. Hence, it should not be leveled in order to cast doubt on the novelty of the present manuscript.

iv) “In most of the situations correlations networks are neither simply random (due to the physical dynamics creating some “structure”) nor perfectly regular (unless restricted to highly symmetric systems and states). Therefore the emergence of complex networks is not a breakthrough by itself.” The reviewer’s point here suggests a misunderstanding of our findings and ignores the fact that we are not claiming the first instance of generally-structured correlations in a quantum simulation experiment, but rather, the first instance *and quantitative characterization* of generated correlations of a very specific (i.e., small-world) and significant structure in a quantum simulation experiment. We reiterate, these specifically-structured correlations were observed and characterized on an actual quantum computer, which stands in contrast to their prior theoretical prediction derived from simulations on a classical computer. It is true that unitary simulation of many quantum systems will yield structured correlations *to some degree* that are neither fully random nor fully regular. However, one of the main results of the theoretical work prior to this manuscript was that, at least in the context of one-dimensional quantum cellular automata, different dynamical systems (i.e., different QCA update rules) lead to varying degrees of structure forming in their correlations [L. E. Hillberry et al., 2021, doi:<https://doi.org/10.1088/2058-9565/ac1c41>]. For instance, while rules T_1 , T_{14} , and F_4 (a one-dimensional five-site rule) also exhibit a significant degree of complex mutual information network structure, the T_6 rule was found to generate the most among the rules studied. Meanwhile, it may indeed be the case that some subset of general quantum simulation experiments display at least as much generated mutual information network complexity as the T_6 rule. It would be fascinating to observe such results, though it is beyond the scope of this work. This however, calls into focus one of the contributions of our work to the field: we have furnished herein both an experimental and analytic methodology for investigating such questions in the future. Hence, we are not simply claiming qualitatively to have generated *some* correlation network structure, which we agree would be a relatively weak claim given the general presence of correlation structure in dynamical quantum simulation. Rather, we have chosen to simulate a particular model– the T_6 rule, which was shown to generate more significant complexity than other T_R rules given fixed system parameters (initial conditions, chain length, etc.)– and thereby established *quantitatively*, via rigorous complex network analysis, the generation of dynamically fluctuating small-world networks. Especially since this was achieved in the face of quantum processor noise, it does constitute a breakthrough.

- 3) Experimental classical correlation network. The experimental implementation in the Sycamore platform is the major novelty of this work. Chains between few and more than 20 qubits

are considered. Looking at the dynamics of clustering and path length for the classical mutual information matrix (Fig.3), a temporal window (between 4 and 12 cycles) is observed in which both quantifiers agree in the experimental post-selected and in the theoretical prediction, while in absence of post-selection the state becomes fully mixed. Post selection is indeed needed to get a mutual information network. The price to pay is the reduced number of selected runs out of the total, a very short time window of agreement before reaching a state completely determined by post selection, and access to only classical mutual information. Actually post-selection by itself is apparently enough to induce complex networks even starting from uniform random states. This manifests in the growing clustering (dashed line in Fig.4 a), rather significant for these states even if smaller than for the T6 post-selected ones.

We concur with the overarching statement, **“The experimental implementation in the Sycamore platform is the major novelty of this work.”** Despite the drawbacks to our approach highlighted by the reviewer (short coherence window, access only to classical mutual information, etc.), which are mainly the consequence of the current developmental stage of digital quantum (and not just superconducting) processors, it is important to note that prior to our work, no such simulation of a quantum cellular automaton on a digital quantum computer was successfully demonstrated anywhere in literature. Hence, our experiment has broken entirely new ground and demonstrated that a wholly new class of systems is simulable, albeit imperfectly, on current generation quantum computers. The techniques demonstrated here are directly adaptable to other QCA and larger systems as hardware fidelity improves. We address additional points individually below.

i) **“Chains between few and more than 20 qubits are considered. Looking at the dynamics of clustering and path length for the classical mutual information matrix (Fig.3), a temporal window (between 4 and 12 cycles) is observed in which both quantifiers agree in the experimental post-selected and in the theoretical prediction, while in absence of post-selection the state becomes fully mixed. Post selection is indeed needed to get a mutual information network. The price to pay is the reduced number of selected runs out of the total, a very short time window of agreement before reaching a state completely determined by post selection, and access to only classical mutual information.”** It is correct to point out that post-selection is one of the primary error mitigation techniques employed in this work in order to enable the titular experimental demonstration. We would like to contextualize what the attendant “price to pay” constitutes. First, the reduced number of selected runs out of the total does reduce the calculation fidelity of the various observables (population, clustering, etc.) roughly according to shot noise Poisson statistics. However, this degradation in fidelity is quadratically slower than the rate at which post-selection mitigates errors in the raw experimental data both as a function of QCA cycle depth and system size. Second, the end of the “very short time window of agreement” occurs at QCA cycle $t = 12$, independent of system size. While this may appear to be a very short coherence window in terms of QCA cycle depth, simulating our largest system— 23 qubits— out to $t = 12$ corresponds to a two-qubit gate volume of $1,056 \sqrt{i\text{SWAP}}^\dagger$ gates. This constitutes a very high gate count for any current-generation NISQ processor and thus may be regarded as a novel accomplishment in and of itself. Moreover, the size-independent scaling of the decoherence cycle depth under post-selection is favorable for subsequent QCA simulation experiments, since improvements in hardware error rates will directly translate into deeper (more cycles irrespective of system size) or wider (more qubits irrespective of cycle depth) circuits accessible by experiment. Finally, despite the fact that post-selection

only allows access to computational z -basis measurements, and thus classical (Shannon) mutual information, we show in Supplementary Information (see SI Fig. 10) that Shannon mutual information acts as a reliable proxy for quantum (von Neumann) mutual information in the calculation of mutual information complex network observables. In other words, while Shannon mutual information will not capture all of the correlations extant in the T_6 QCA dynamics, it detects a sufficient quantity so as to be able to draw conclusions regarding the complexity of the underlying quantum state. Ours, of course, is not the first publication to have used Shannon mutual information to infer the existence of physics more accurately embodied in von Neumann mutual information. For instance, [F. C. Alcaraz and M. A. Rajabpour, 2013, doi:<https://doi.org/10.1103/PhysRevLett.111.017201>] employed Shannon mutual information (as a proxy for von Neumann mutual information) between bi-partitions in order to back out universality classes of quantum spin chains (e.g., the XXZ model) near critical points. Beyond the topic of mutual information specifically, it is well-known that “classical” measurements in a single basis can be used effectively to infer characteristics of an underlying quantum state. The distribution of classical bit strings in the computational z -basis output of random quantum circuits, for example, exhibits Porter-Thomas structure while the corresponding output of noise-dominated dynamics exhibits an incoherent uniform random structure [F. Arute et al., 2019, doi:<https://doi.org/10.1038/s41586-019-1666-5>]. Such measurements are therefore able to delineate between a highly-correlated state and a completely uncorrelated state despite the fact that no explicit measure of quantum correlations, such as von Neumann mutual information, is being constructed to do so.

ii) “Actually post-selection by itself is apparently enough to induce complex networks even starting from uniform random states. This manifests in the growing clustering (dashed line in Fig.4 a), rather significant for these states even if smaller than for the T6 post-selected ones.” While it is true that post-selected incoherent uniform randomness does generate a clustering coefficient that grows with system size (Fig. 4a), similar to the post-selected experimental QCA data, and a path length metric that tracks post-selected experimental QCA data closely (Fig. 4b), the node strength distribution (Fig. 4c) of post-selected randomness is distinctly narrower than that of both the post-selected experimental QCA data and emulated QCA data. This indicates a relative deficiency in both the very highly-connected nodes and very sparsely connected nodes that are strong indicators of “small-worldness” in a network. Moreover, as can be seen in Fig. 2, post-selected randomness (from $t \approx 15$ onward) fails to accurately reproduce the emulated population dynamics of the T_6 QCA. Hence, while post-selected randomness may produce a reasonable facsimile of a small-world network, it neither does so to the extent that our post-selected experimental data does, nor does it do so in a manner that accurately follows system dynamics.

It would also be worth to comment on the platform choice, given the limitations imposed by decoherence, spoiling any correlations in few cycles and allowing to observe only classical correlations through post-selection. The authors could comment on the choice of this platform and on which performance could be expected in an atomic one, as envisaged in other works (e.g. Rydberg atoms).

QCA are digital and run in discrete time on quantum circuits. They are not non-equilibrium dynamics simulations of Hamiltonians. The reviewer’s referencing of Rydberg chains is an implicit allusion to the latter. In fact, digital quantum computers of any reasonable scale on any platform are extremely difficult to obtain access to. We thus feel this is an unreasonable

criticism by the reviewer. Yes, IonQ’s future digital quantum computer with trapped ions might be a better choice, due to higher fidelity – there will always be a better experiment later, but that’s some years down the line. We will now be very specific on these points, as the referee may possibly not be aware of the limitations of current digital quantum processors.

First, we justify the use of a universal, gate-model processor against an analog quantum simulator. Chains of Rydberg atoms up to 51 and trapped ions up to 53 sites long have been used simulate the many-body dynamics of kinetically-constrained (sometimes called PXP [C. J. Turner, 2018, doi:<https://link.aps.org/doi/10.1103/PhysRevB.98.155134>]) and quantum Ising-type spin models out to microseconds and milliseconds long, respectively [H. Bernien et al., 2017, doi:<https://doi.org/10.1038/nature24622>] [J. Zhang, 2017, doi:<https://doi.org/10.1038/nature24654>]. These are the existing experiments we know of that most closely approximate ours on the analog simulation side of the field. While these experiments exhibit larger qubit counts and in some instances better coherence properties, analog quantum simulators are highly-constrained in the systems they can simulate by the native degrees of freedom and interactions of the underlying simulator. That is, while particular analog simulation platforms may be appropriate for simulating particular quantum cellular automata, it would be very difficult to design one as a general-purpose QCA simulator. This, of course, reflects the general need for universal, gate-model machines writ large. As such, though it is required in our experiment to combat higher noise levels than those found in analog simulators through a variety of techniques, some of which, it is true, do limit the observables we are able to calculate, our approach to QCA simulation is generalizable not just to other QCA in one-dimension, but to those of higher dimension and greater connectivity as well, given sufficiently improved noise characteristics of the underlying quantum processor.

Second, having addressed the issue of analog versus digital quantum simulation, we remark upon our particular choice of digital platform. There currently exist various approaches to constructing gate-model quantum computers, among them superconducting, trapped ion, neutral atom, photonic, and topological qubit-based processors. Each of these approaches involves design trade-offs. At the time of performing our experiment, the large majority of cloud-accessible gate-model quantum computers were either trapped ion (e.g., Quantinuum, IonQ) or superconducting machines (e.g., Google, Rigetti, IBM). Though prosaic, it is worth noting that for researchers who wish to perform quantum simulation studies on quantum hardware and who do not themselves maintain an experimental platform, such cloud-accessible machines remain the best option. The relevant question then is, “Why use Google’s superconducting processors?” The capability of a given quantum processor involves trade-offs in terms of qubit count, qubit connectivity, decoherence times (relative to gate execution times), native gate expressiveness, and gate and state-preparation and measurement error rates. While trapped ion two-qubit error rates can be somewhat lower than superconducting error rates– e.g., $\sim 0.3\%$ for Quantinuum’s H1.1 processor– and are fully-connected due to the ability to shuttle ions, cloud accessible trapped ion processors remain rather size-limited with respect to superconducting processors [T. Lubinski et al., 2021, doi:<https://doi.org/10.48550/arXiv.2110.03137>]. For example, at the time our experiment was performed, and up until very recently, Quantinuum’s cloud accessible processor could only load up to 12 ions and IonQ’s processor could only load up to 11 ions. By comparison, we have simulated QCA as large as 23 qubits long. Moreover, we did not require all-to-all connectivity for our QCA circuits since all interactions were nearest-neighbor. We also note that our employment of the Floquet calibration technique allowed for the additional suppression of two-qubit gate error in our system beyond the $\sim 1.4\%$ rate determined by cross-entropy benchmarking. As for superconducting processors, the Rainbow and Weber processors on which we ran our experiments are at least as performant as those offered by other superconducting processor vendors such as IBM or Rigetti. In summary, it is unlikely that run-

ning our QCA circuits on a different current-generation, gate-model quantum processor would have resulted in a positive step-change in terms of fidelity, although we would be fascinated to see our results repeated or improved upon on a different architecture.

In conclusion, the manuscript is overall well presented and sound, the topic and the experimental realization are interesting and original and it deserves to be published. On the other hand, Nature Communication is rather selective. The main novelty of this work is the experimental detection of a classical network of correlations in post-selected states of a QCA. Post selection represents a strong limitation in the sense that actually even post-selected incoherent uniform random states would lead to a clustering of the order of (even if smaller than) the experimentally reported one. I am therefore recommending this paper for a more specialized journal.

We address the reviewer’s final comment in a point-by-point manner below.

i) **“In conclusion, the manuscript is overall well presented and sound, the topic and the experimental realization are interesting and original and it deserves to be published.”** We thank the reviewer for their generally favorable review of our work.

ii) **“On the other hand, Nature Communication is rather selective.”** Given that the reviewer is of the opinion that “the manuscript is overall well presented and sound, the topic and the experimental realization are interesting and original and it deserves to be published,” it is difficult to understand what deficiencies our manuscript should need to rectify— other than the reviewer’s perceived lack of novelty— in order to be considered equal to Nature Communications’ selectivity.

iii) **“The main novelty of this work is the experimental detection of a classical network of correlations in post-selected states of a QCA.”** As already noted, this is an incomplete characterization of our work. We do not simply detect the existence of some arbitrary classical network of correlations. We show the dynamical generation of a network of correlations, using a classical mutual information metric as a well-justified proxy for quantum mutual information, with a particular (i.e., small-world) structure that is of a special status in complex network science and which describes many networks that occur in the natural world. Given the evidence in Supplementary Information Fig. 10, the Shannon mutual information metric employed is sufficiently effective as a proxy for von Neumann mutual information that it infers the existence of a small-world designation also for the quantum correlations in the system. As such, we not only demonstrate the first digital quantum simulation of a quantum cellular automata, which exhibits coherent dynamics, we also show that dynamical quantum systems can generate physical complexity— in this instance, of the small-world network variety— even in the face of substantial environmental noise, which post-selection mitigates but does not completely remove. Since the QCA simulated here furnish simplified computational models for how quantum complexity emerges in the real world, similar to how classical cellular automata do for classical complexity, our results suggest that physical complexity arising in natural quantum systems may also have the ability to resist certain levels of environmental noise.

iv) **“Post selection represents a strong limitation in the sense that actually even post-selected incoherent uniform random states would lead to a clustering of the order of (even if smaller than) the experimentally reported one.”** While it is true that environmental noise requires us to use post-selection, the small “gap” that the reviewer observes between the post-selected experimental clustering and the post-selected incoherent uniform random clustering in, say, Fig. 4a is emphatically not a result of the use of Shannon, rather

than von Neumann, mutual information. Put another way, suppose we were in possession of a fault-tolerant quantum computer with exponentially suppressed error rates. In this instance, we would have no need for post-selection as an error mitigation tactic and, as a result, we could present our complex network results using von Neumann mutual information. *However*, we still could use post-selected incoherent uniform randomness as a baseline against which to plot our complex network metrics- nothing would prevent us from doing so. The resulting time-dependent dynamical clustering plots would look nearly identical to the gold curves in Supplementary Fig. 10a, which we also emphasize overlay the blue curves (Shannon MI) to a very good degree of accuracy. The point here is that post-selecting on the incoherent uniform random state will lead to non-trivial clustering, on the order of the exact, emulated clustering, even if post-selection is not required in order to mitigate errors in the experimental data. Therefore, it is not a shortcoming in our experimental methodology that leads to such a small clustering gap, as the reviewer appears to imply, but rather, it is simply a fact of the system at hand that the underlying symmetry of the T_6 rule evidently leads to a large, average clustering for this particular initial condition. But that is not to say that there is no worth in performing the time-dependent quantum simulation of the T_6 rule in favor of post-selecting the incoherent uniform random state and plotting those complex network metrics instead. For one, such a method does not produce a dynamical small-world network, which the T_6 rule does. Moreover, other complex network measures such as node strength distribution (Fig. 4c) also show a clear deficiency in the small-world character of post-selected randomness against the T_6 small-world character. Hence, the correct interpretation of our data is: given that the \mathcal{O} -symmetry of the T_6 rule already forms a state with some degree of small-world correlations when applied to incoherent uniform randomness, any signature of additional small-world correlations that build up due to the T_6 dynamics, while correctly replicating local observables like population, is a significant result. This is precisely what we have shown.

v) **“I am therefore recommending this paper for a more specialized journal.”** This recommendation appears to conflate the issue of novelty with specialization. The majority of the reviewer’s objections to our manuscript revolve around a perceived lack of novelty in the work. However, the recommendation to publish in a more specialized journal has more to do with how constrained the potential scope of impact of the work is to a particular field and less to do with the degree of novelty within the relevant field or fields covered by the topic at hand. Having argued for the novelty of our work in previous points above, we believe the suggestion to constrain our work to a more specialized journal is inappropriate for at least two reasons. First, this manuscript is highly-relevant and broadly applicable both for the quantum computing and information communities (error mitigation techniques, quantum applications, etc.) and the complexity science/cellular automata communities. It thus belongs in a manifestly multi-disciplinary journal, i.e., Nature Communications. And second, as it is the first experimental simulation of quantum cellular automata on a digital quantum computer anywhere, it belongs in a journal with wide-enough reach to herald the feasibility of these types of experiments going forward. To consign our results to a more specialized journal would be to deny, for the time being, the potential of digital quantum simulation and complexity science to inform one another in a highly visible and impactful manner. Such a denial, in our view, would hamper the advancement of both fields.

Reviewer #3 (Remarks to the Author):

This paper describes an experimental study of quantum cellular automata (QCA) using superconducting qubits. QCA can produce complex quantum states using only geometrically-local

operations applied repeatedly. The authors realize a particular QCA comprising controlled-Hadamard gates, and show that the observed evolution in the computational basis agrees quite well with noiseless simulations. This agreement depends critically on post-selection, where measurement results that do not respect a known symmetry of the QCA, due to errors, are discarded. The authors then measure 2-site correlations in the computational basis between all pairs of qubits, and construct matrices of the resulting Shannon mutual information. Interpreting these as graph adjacency matrices, they analyze the correlations in terms of various quantities which arise in the study of complex networks. From these they conclude that the QCA is indeed creating complex quantum states from simple operations.

I find this paper well-written and quite elegant. As far as I can tell, the experimental methods and the analyses are generally sound. More broadly, the motivation for studying QCA is explained in a compelling way, and the experimental results look quite good. Unfortunately, I am not qualified to comment on the relation of this paper to previous work on QCA or network analysis, nor on the gate calibrations or prospects for classical simulation mentioned at the end. I trust that the other referees will be able to vet these aspects in more detail.

While I think this is nice paper, I am on the fence about whether it should be published in this journal or in a more specialized one. I have a few major concerns, as well as some more minor ones.

We thank the reviewer for a careful reading of our manuscript and appreciate the opportunity to address their concerns and comments below.

Major concerns:

- The authors claim that this is the first experimental realization of QCA on a digital quantum processor. However, any repeated application of identical 1- and 2-qubit gates seems to fit the broad definition of QCA given in the main text. For instance, a Trotterized simulation of a 2-local Hamiltonian (e.g., of a transverse-field Ising model or a Heisenberg model in 1D) seems very similar to, if not equivalent to, a QCA. Trotter circuits have certainly been realized before, and also generate complex quantum states using repeated local operations. There is nothing exotic about that, as far as I can tell. Besides the interpretation, where exactly lies the boundary between this experiment and earlier ones with a similar structure?

It is an interesting point raised by the reviewer. Our claim is certainly not that we have performed the first Trotterized time evolution of a model Hamiltonian on a digital processor, many examples of which already exist in the literature, e.g., [A. Smith et al., 2019, doi:<https://doi.org/10.1038/s41534-019-0217-0>] and [F. Arute et al., 2020, <https://arxiv.org/abs/2010.07965>]. In contrast, we demonstrate the first experimental realization of a discrete time QCA on a digital quantum processor. It is true that our discrete time QCA can be approximated as arising from the low-order Trotterized evolution of *some* Hamiltonian. However, there are distinct aspects in which the simulation of QCA differs from the general activity of Trotterized Hamiltonian evolution. Hence, we append the following to the “Quantum Cellular Automata in 1D” section in Supplementary Material, highlighted in yellow:

“The continuous time versions of the QCA considered herein, of which the T_6 rule is a specific instance, evolve according to *kinetically-constrained* Hamiltonians. Models of this type were originally introduced to study the dynamics of glassy systems [F. Ritort and P. Sollich, 2003, <https://arxiv.org/abs/cond-mat/0210382>]. Their characteristic attribute is that degrees of freedom evolve explicitly as a function of the state of their

neighborhood— this can be seen as embodied via the projection operators in Eq. 1. Given that kinetically-constrained models were originally introduced to study glassy dynamics, it is a reasonable question as to whether the T_6 QCA that we simulate corresponds to some spin glass-related model that has already been simulated on a gate-model quantum processor. As demonstrated in Section V.A. of [L. Hillberry, 2020, doi:<https://doi.org/10.1088/2058-9565/ac1c41>], the continuous time, *analog* T_6 QCA can be *approximated* by a transverse-field Ising model (TFIM) with very particular structure. However, there are a few, very important ways in which the discrete time simulation of rule T_6 differs from (and therefore is not simply equivalent to) Trotterized evolution of the TFIM. First, while the discrete time QCA cycle unitary $U(T_6; t, t + 1)$ commutes with the dynamical invariant operator used for post-selection, \mathcal{O} , the TFIM Hamiltonian does not. Hence, the two systems have different conservation laws, although for certain discrete time steps it is the case that the TFIM evolution unitary does commute with \mathcal{O} [V. Subrahmanyam, 2003, doi:<https://doi.org/10.1103/PhysRevB.68.212407>]. Second, using the TFIM to model the dynamics of the continuous time T_6 rule only works when the Hermitian activation generator is the Pauli X operator. However, our activation unitary is the Hadamard operator, $V_j = H$, which cannot be generated by X alone. Finally, even if some slight alteration of the TFIM could generate the local Hadamard activation unitary, say via $V_j = \exp(-i\alpha H)$, this would correspond to a Trotter step of $\alpha = \pi/2$, which cannot be regarded as an accurate Trotterization of the continuous limit. Therefore, the discrete time T_6 QCA considered in this work cannot simply be mapped onto the TFIM. Since we have ruled out the TFIM as the appropriate Trotterized model Hamiltonian corresponding to the discrete time T_6 QCA, and since to our knowledge no other kinetically-constrained Hamiltonian has been simulated on a digital quantum processor, we believe the present work to constitute the first quantum simulation of a QCA on a gate-model quantum computer.”

Of course, it may be the case that one of the discrete time QCA in the T_R family of rules, such as the T_6 rule, might eventually find an accurate representation as the Trotterized limit of some new, kinetically-unconstrained model Hamiltonian (a theoretical endeavor beyond the scope of this work). However, such a discovery would not detract from the novelty of our demonstration. In short, just because *some* Hamiltonians (or Floquet systems for that matter) have been simulated on gate-model processors, such as Heisenberg chains and Fermi-Hubbard Hamiltonians, it doesn’t mean that quantum simulation of new quantum systems, such as kinetically-constrained QCA, should be wholly abandoned, for that would preclude the potential discovery of new physics— exactly the purpose for which quantum computers were originally intended. Moreover, while complex quantum states doubtlessly arise in the Trotterized evolution of other Hamiltonian systems, our manuscript presents the first characterization of the complexity of many-body quantum states using tools from complex network theory in the face of digital quantum processor noise.

Finally, regarding previous analog quantum simulation of QCA, in the “Quantum Cellular Automata in 1D” section of Supplementary Information we write: “Another important totalistic rule is T_1 , also called the PXP model in many-body quantum literature when run in continuous time [3];” The PXP Hamiltonian has been analog-simulated with Rydberg atoms. The lowest-order Trotterization of the Rydberg PXP Hamiltonian is nearly exactly what we have defined as QCA rule T_1 , the exception being that the Hadamard gate is replaced with the gate $\exp(i dt X)$ where X is the Pauli- X operator and $dt \ll 1$ is the Trotter time step.

- It seems reasonable enough to interpret 2-qubit correlations as describing a graph. However, there’s a conceptual leap made in the second half of the paper wherein the system starts to get treated like an actual network on this basis. The authors rely heavily on this analogy, and use it to motivate several figures of merit, including clustering, shortest path length, and node strength distribution. In the supplemental material these quantities are explicitly explained in terms of transportation networks. But the underlying quantity in these experiments is not a transportation network or the like; it is simply 2-point correlations between a set of binary random variables. It is not obvious to me that these network-inspired figures of merit are appropriate/meaningful for analyzing such correlations, and for concluding that the underlying states are complex. In fact, the matrices in Figs. 4d-g don’t look particularly exotic until they’re turned into seemingly-abstruse figures of merit. Have these network-inspired quantities been used to study correlations in other papers (quantum or otherwise)? The authors should explain in more detail why a reader should care about the quantities plotted in Figs. 3 and 4.

We understand the reviewer’s hesitancy with respect to the use of complex network measures in analyzing the correlation physics of an experimental quantum system. We endeavor to address this hesitancy in a point-by-point manner below.

1) **“It seems reasonable enough to interpret 2-qubit correlations as describing a graph. However, there’s a conceptual leap made in the second half of the paper wherein the system starts to get treated like an actual network on this basis. The authors rely heavily on this analogy, and use it to motivate several figures of merit, including clustering, shortest path length, and node strength distribution.”** The system under study is a one-dimensional chain of qubits evolving according to the T_6 QCA rule. We emphasize that we use both one-point (population) and two-point (Shannon mutual information) measures as probes of the system. The population observable furnishes an intuitive visualization of non-equilibrating QCA dynamics. Meanwhile, it is important to point out that Shannon mutual information between all pairs of qubits *is* a network, and is not merely treated as one. It is a network of correlations between the L nodes (qubits) in the system. As such, it is appropriate to use the well-established tools of complex network science in order to analyze the properties of this network. This is especially true since the question at hand is: “Can a current-generation digital quantum computer generate emergent physical complexity, of the type encountered in complex network and complex systems science, even in the face of noise?”. There is no better set of tools to answer this question than those adopted from complex network/systems science and no more natural observable to predicate the analysis on than an accessible measure of two-qubit correlations. As described elsewhere, we use Shannon mutual information in particular due to the necessity of performing post-selection on our data.

2) **“In the supplemental material these quantities are explicitly explained in terms of transportation networks. But the underlying quantity in these experiments is not a transportation network or the like; it is simply 2-point correlations between a set of binary random variables. It is not obvious to me that these network-inspired figures of merit are appropriate/meaningful for analyzing such correlations, and for concluding that the underlying states are complex.”** The transportation network analogy is used pedagogically in the Supplementary Information for readers who are unfamiliar with complex network science, because of its simple interpretation. We place the following in the “Complex Network Measures” section of Supplementary Information (highlighted in yellow) in order to more completely

contextualize our use of these complex network measures:

“In wireless communications networks [R. Urgaonkar and M. J. Neely, 2012, <https://ieeexplore.ieee.org/document/6311231>], where nodes are connected by channels with limited bandwidth, or brain networks, where functional connectivity is distinct from spatial connectivity [E. Bullmore and O. Sporns, 2009, doi:<https://doi.org/10.1038/nrn2575>], network measures carry similar meaning as the mutual information networks herein. For instance, a communications network with higher channel capacity (upper bound on amount of information transmissible between two nodes) should have a shorter path length between the nodes, because it is easier for information to move between the nodes. Similarly, in a functional brain network described by mutual information, nodes (i.e., functional regions) within a pair that share more mutual information will affect one another’s state to a greater extent. Hence, path length should also be lower in this instance. The intuition behind these examples is that thresholding a network with a minimum channel capacity or mutual information will only leave edges between strongly-linked nodes. Weak links will be removed. Hence, if only weak links exist between nodes prior to thresholding, they will be removed, contributing to a longer path length— which after thresholding is calculated just by counting the number of edges traversed— between the nodes. Similar, common meanings also exist for clustering and node strength distribution between QCA, communications, and functional brain networks.”

Therefore, the issue of determining whether or not the mutual information-based complex network analysis in the present work is appropriate for determining whether the QCA states and dynamics generated are complex, in some sense, boils down to assessing whether or not similar analyses performed for functional brain and communications networks were well-justified. However, invalidating *those* analyses would appear to be a difficult task, given their widespread employment and acceptance in the literature— see again [R. Urgaonkar and M. J. Neely, 2012, <https://ieeexplore.ieee.org/document/6311231>] and [E. Bullmore and O. Sporns, 2009, doi:<https://doi.org/10.1038/nrn2575>]. In addition, we reiterate that it is not only our complex-network analysis that supports our conclusion that the T_6 QCA dynamics are complex— the population dynamics also support this fact.

3) “In fact, the matrices in Figs. 4d-g don’t look particularly exotic until they’re turned into seemingly-abstruse figures of merit.” Examining a network’s adjacency matrix is at best a qualitative assessment of any structure, and the designation thereof as “exotic” or not is highly subjective. The direct network visualizations do somewhat better, but the complex network measures are well-established quantitative tools and the most quantitatively appropriate for the task of uncovering physical complexity in a mutual information network. We include both the adjacency matrix and network visualizations even though they portray the same data because the former is possibly more familiar to the quantum information and quantum many-body communities while the latter is more familiar to the network science and complexity communities. As remarked upon both in the previous and subsequent points, far from being abstruse, the mutual information-based complex network measures we use in this publication are well-established analytic tools that have been applied to mutual information networks across a variety of scientific domains. This includes quantum systems (see below).

4) “Have these network-inspired quantities been used to study correlations in other papers (quantum or otherwise)?” In the quantum setting, network measures evaluated on mutual information adjacency matrices accurately locate quantum phase transitions [M. Valdez, 2017, doi:<https://doi.org/10.1103/PhysRevLett.119.225301>]. In the classical setting, network measures evaluated on mutual information adjacency

matrices map functional correlations in brain activity [Z. Wang, 2009, doi:<https://doi.org/10.1186/1475-925X-8-9>]

5) “The authors should explain in more detail why a reader should care about the quantities plotted in Figs. 3 and 4.” One-point observables are conveniently visualized as spacetime grids (Fig. 2a). Two-point correlations, while a typical metric in many fields, are more challenging to visualize. Interpreting the matrix of two-point correlations as a network and subsequently evaluating scalars (network measures) designed to elucidate network structure compresses the correlation data into fewer variables. In order to summarize points 1-5 addressed here, we append the following (highlighted in yellow) to the second paragraph in the “Mutual information network analysis” section of our manuscript:

“While transportation networks provide an intuitive interpretation of these complex network measures, we emphasize that clustering, path length, and node strength distribution have seen widespread use in analyzing the structure of mutual information networks and in drawing conclusions regarding the physical complexity of the underlying system in both classical and quantum systems. For example, applying these measures to mutual information networks derived from electroencephalographic or fMRI data has been used to elucidate structure-function correlations in the brain [E. Bullmore and O. Sporns, 2009, doi:<https://doi.org/10.1038/nrn2575>] [Z. Wang, 2009, doi:<https://doi.org/10.1186/1475-925X-8-9>]. In addition, the measures have been used along with earthquake time series-derived mutual information to model seismicity [A. Jiménez, 2013, doi:<https://doi.org/10.1016/j.physa.2013.01.062>] and with mutual information in wireless networks to explain routing efficiency [R. Urgaonkar and M. J. Neely, 2012, <https://ieeexplore.ieee.org/document/6311231>]. Finally, complex network measures calculated on quantum mutual information networks in quantum Ising and Bose-Hubbard models are able to detect quantum phase transition critical points [M. Valdez, 2017, doi:<https://doi.org/10.1103/PhysRevLett.119.225301>]. Hence, the use of clustering, path length, and node strength distribution in conjunction with mutual information is a well-established, quantitative procedure with predictive power. We employ this procedure in order to understand the structure of correlations in our QCA circuits and to observe the emergence of physical complexity in the presence of quantum processor noise.”

- On a similar note, I’m uneasy about the use of classical, as opposed to quantum, mutual information throughout. I understand that it would probably not be feasible to measure the quantum mutual information, and I am not suggesting that this is essential. But I’m worried that the classical mutual information could depend heavily on the measurement basis. It would be reassuring to see, even if just in a small simulation (no need for further experiments), that the same conclusions follow from local measurements along the x- or y-axes, not just the z-axis.

The reviewer is correct regarding the infeasibility of experimentally measuring (e.g., von Neumann) quantum mutual information with any degree of accuracy past one or two QCA cycles, which is not long enough for a complex network to emerge. This is because construction of quantum mutual information requires measurements of two-point correlators that are non-diagonal in the z -basis, for example $\langle X_i Y_j \rangle$ — see Eq. 25 in Supplementary Information. The required non-diagonal measurements cannot generally be calculated in an error-mitigated fashion through post-selection since in general it is not the case that $[\sigma_i^\mu \sigma_j^\nu, \mathcal{O}] = 0$, where \mathcal{O} is the dynamical invariant defined in Eq. 10 of Supplementary Information. Therefore, we focus on Shannon mutual information in the main text, because it can be constructed solely from measurements in the computational z -basis, which

are in turn amenable to post-selection.

The main point to address is therefore whether or not Shannon classical mutual information acts as a reliable proxy for von Neumann quantum mutual information, indicating the formation of complex networks (as measured by clustering, path length, and node strength distribution) when quantum mutual information would have in the experimental setting. In order to resolve this point, we direct the reviewer’s attention to Figs. 10-11 in Supplementary Information and their corresponding discussion, highlighted in yellow:

“As validation for this choice, we numerically emulate $L = 19$ qubits initialized with a single-bit flip and evolving under T_6 for 10,000 cycles. At each cycle we calculate the relative distance between Shannon- and von Neumann-based network measure, $f \in \{\mathcal{C}, \ell, g_i\}$, as $[f(I) - f(I^{\text{vN}})]/f(I^{\text{vN}})$ and relative Frobenius distance between the two mutual information measures $\|I - I^{\text{vN}}\|_{\text{F}}/\|I\|_{\text{F}}$, where the Frobenius norm of a matrix M with elements M_{ij} is defined as

$$\|M\|_{\text{F}} = \sqrt{\sum_{i,j=0}^{L-1} |M_{ij}|^2}. \quad (1)$$

Fig. 10a shows a direct comparison of the Shannon- (blue) and von Neumann- (gold) based mutual information network clustering over the first 100 QCA cycles for $L = 19$. Dashed lines represent the corresponding clustering values for post-selected incoherent uniform randomness. Visually, it is clear that the two different mutual information metrics lead to very similar clustering values both for the emulated QCA and post-selected incoherent uniform randomness. In particular, the slight shifts that are observed are insufficient to erase the existence of the coherence windows between $t \sim 4$ and $t \sim 12$ established in Fig. 3 in the main text. Fig. 10b quantifies and extends this observation with respect to all three complex network measures discussed in the main text: it shows the distribution of relative differences between Shannon- and von Neumann-based mutual information network measures across 10,000 QCA cycles for $L = 19$. Histogram curves represent QCA data and vertical dashed lines represent the corresponding differences for post-selected incoherent uniform randomness. Clustering is the most similar between Shannon- and von Neumann-based mutual information (blue histogram, mean relative difference 0.8%, median 0.7%). Path length (gold histogram) exhibits a mean relative difference of -38% , (median -21%) while that of node strength (green histogram) is 12% (median 7%). For the $L = 19$ post-selected incoherent uniform random state, the relative difference Shannon- and von Neumann-based clustering is -6.1% while that of path length is 10% and average node strength is -14% . Fig. 10c shows that as a function of system size, the mean relative differences between network measures over 10,000 cycles (solid lines) and post selected incoherent uniform random states (dashed lines) tend towards zero roughly as a power law. Colors are as in panel b. The 10,000 cycle emulations terminate at $L = 19$ due to computational cost. Hence, we conclude not only that Shannon mutual information serves as an effective proxy for von Neumann mutual information in the calculation of complex network measures in our regime of interest, $L \in \{5, 7, \dots, 23\}$, but that it would become an even more effective proxy in the large L limit. As in Fig. 10b-c, but instead of relative differences of network measures, Fig. 10d-e shows the relative Frobenius distance between Shannon- and von Neumann-based mutual information matrices directly. The mean relative Frobenius distance over 10,000 QCA cycles is 10% (median 9%) while that of post-selected randomness is 14% . From the above analysis, we conclude that Shannon mutual information is a reliable proxy for von Neumann mutual information at the level of a few to several percent for $L = 19$ and that the relative difference between the two

types of mutual information tends monotonically towards zero as L increases.

Finally, aside from the requirement to perform post-selection for error mitigation purposes using the operator \mathcal{O} in Eq. 10, we justify the use of the computational z -basis as the relevant measurement basis in which to construct the Shannon mutual information as opposed to the y -basis or x -basis (which, for instance, is related to the z -basis by L -qubit Hadamard transform). First, we note that measurement in the y -basis would yield no Shannon mutual information between any pairs of qubits for the initial conditions considered in this work. This is because all qubits are either initialized in the $|0\rangle$ or $|1\rangle$ state and a series of Hadamard activation unitaries will never move the dynamics of states so initialized out of the xz -plane of a qubit’s Bloch sphere. It remains to show that the choice of z -basis is at least as sensible as choosing the x -basis. Fig. 11 shows the clustering coefficient for Shannon mutual information networks calculated based on measurements in the x -basis (blue curves) and z -basis (orange curves) as a function of QCA cycle for three different system sizes, $L \in \{15, 17, 19\}$. While the z -basis Shannon mutual information network clustering tracks the von Neumann mutual information network clustering (Fig. 10a) closely, the x -basis Shannon mutual information network clustering fluctuates only slightly from the Shannon mutual information network clustering of incoherent uniform randomness (black dotted line), which is a basis-independent quantity, for the system sizes shown. This is intuitive, since the amount of correlation structure (in this case clustering) between degrees of freedom in one measurement basis comes at the expense of correlation structure carried in a different basis if the total degree of correlations is bounded by the von Neumann mutual information. Taken together with the rest of the evidence presented in Fig. 10, it is clear that z -basis Shannon mutual information is the correct network quantity with which to establish the existence of correlations with small-world structure.”

Given this discussion, the reviewer is correct in suspecting that the amount of Shannon mutual information shared between qubits might depend upon the measurement basis. However, rather than our results being invariant with respect to measurement basis, there exists a naturally-preferred measurement basis– the z -basis– for which the structure in the Shannon mutual information network most closely resembles the corresponding structure in the associated von Neumann mutual information network. This is as it should be, since the extent to which Shannon mutual information calculated in a particular basis captures the physics of the underlying von Neumann mutual information depends directly upon the extent to which it does so in the other bases.

Minor concerns:

- The authors claim in their conclusion that existing quantum processors can efficiently simulate 1D QCA. Yet, in the supplemental material (SM) they say that the retained count fraction (after post-selection) decays roughly exponentially with the number of qubits for fixed t . In other words, the time required to get a constant number of counts increases exponentially with the system size. How should I reconcile these two statements? Also, it would be helpful to add a plot to the SM showing the same data as in Fig. (SM-7) but with L on the x axis for fixed values of t . (Unless I’ve missed it somewhere.)

We agree with the reviewer that these two statements are contradictory in the usual complexity-theoretic meaning of the phrase “computational efficiency”, which is that simulation time scales polynomially rather than exponentially. This contradiction was a result of imprecise language on our part and we thank the reviewer for pointing it out. In the relevant sentence in the concluding section (Towards beyond-classical QCA): “Here we have

demonstrated that existing quantum processors can efficiently simulate 1D QCA with high fidelity at large gate volume”, we have removed the word “efficiently”. In addition, we add the following to the concluding section (change highlighted in yellow): “(For a more complete discussion of the quantum-simulation complexity of the T_6 rule including post-selection overhead, please see Supplementary Information)”. This comment points the reader to the following discussion, which we place in the “Post-Selection” section of the Supplementary Information:

“Because the combined effects of noise and post-selection require that exponentially-many measurements are discarded as both a function of system size and cycle depth, we consider the effect this has on the computational complexity of simulating the T_6 QCA on the Sycamore-class processors relative to the computational complexity of emulating the T_6 rule using standard classical algorithms. For reference, the main text already provides a qualitative discussion regarding the use of tensor network approaches for emulating QCA. For a general quantum circuit of L qubits and M gates, the Schrödinger emulation algorithm requires $\sim 2^L$ spatial resources and $\sim M2^L$ temporal resources. The Feynman algorithm requires $\sim M + L$ space and $\sim 4^M$ time (Aaronson, 2017). However, the observable \mathcal{O} is also conserved for classical emulation. Therefore, in classical emulations one can use a “domain conserving” basis (see below and Fig. 8) that reduces the Schrödinger requirements to $\sim L^{1.91}$ (space) and $\sim ML^{1.91}$ (time). Although hybrid Schrödinger-Feynman algorithms also exist for quantum emulation, we focus on comparison with the Schrödinger algorithm for simplicity and because the large two-qubit gate count of our largest circuits, $M_{2Q} > 1,000$, makes effective utilization of the Feynman algorithm difficult.

As shown in Fig. 7b, the number of retained counts after post-selection at fixed cycle depth, t , scales as $N_r(t) \sim e^{-b_L(t)L}$ as a function of system size. Since the error, due to shot noise, in calculating a local observable, A , using quantum computer measurements scales as $\langle A \rangle \sim 1/\sqrt{N_r}$, one would need to scale the initial number of counts as $N_c \sim e^{b_L(t)L}$ in order to offset the fraction discarded by post-selection and thus achieve constant precision (We note briefly that we did not follow this methodology in our experimental setup in favor of constant run-time). Therefore, in order to simulate the T_6 QCA on a quantum processor at fixed precision, one would require L spatial resources (qubits) and $\sim d(M(t))e^{b_L(t)L}$ temporal resources. $d(M(t))$ accounts for the fact that gates are often layered in parallel on the processor and that the total gate volume depends upon the desired QCA cycle depth. But the most important element for comparing to the performance of the Schrödinger emulator is the factor $e^{b_L(t)L}$. For the fits shown in Fig. 7b, $b_L(t) \lesssim 0.422$ over the set $t \in \{0, 1, \dots, 13\}$. Hence, $e^{b_L(t)L} \lesssim 1.525^L$. The fit parameters $b_L(t)$ depend upon the error rates in the processor, which will generally improve over time, requiring fewer counts to be discarded at each cycle depth. However, in the absence of fault-tolerant, code-based error correction, the time complexity due to post-selection will remain exponential. In summary, quantum simulation of the T_6 rule obtains a nearly quadratic reduction in spatial computational resources as compared to Schrödinger emulation. In exchange, quantum simulation requires exponential time due to post-selection, while classical time complexity scales polynomially using the Schrödinger algorithm with a domain-conserving basis set.”

We note that our estimate for the scaling in the size bound of the invariant-protected Hilbert space has been revised from $\dim(\mathcal{H}_{L-3}) \sim 1.08^L$ to $\dim(\mathcal{H}_{L-3}) \sim 0.63 L^{1.91}$ due to a refined fitting procedure to the data in Fig. 8. While this refinement reduces the computational complexity of classically emulating the T_6 circuits from exponential to polynomial

space and time complexity (an interesting result in and of itself), we reiterate that our experiment was not intended as a quantum supremacy demonstration and that the discussion surrounding using QCA for beyond-classical demonstrations remains prospective in nature. In order to better reflect the revised Hilbert space scaling and associated classical emulation complexity, we alter the discussion in the concluding section with additional changes highlighted in yellow, among them a framework for recovering exponential scaling in the protected Hilbert space bound: “However, one can recover exponential scaling in the protected Hilbert space by simulating increasingly large chains while fixing the density (rather than number) of initial $|1\rangle$ s well-separated by $|0\rangle$ s. For a fixed density, $\rho_{|1\rangle_s}$, the protected Hilbert space bound scales as the binomial coefficient, $\sim \binom{L}{\rho_{|1\rangle_s L}}$, which asymptotically scales exponentially in L .”

In addition, we concur with and appreciate the reviewer’s suggestion to plot the retained count fraction explicitly as a function of system size for various fixed QCA cycle depths. Please find the new plot in the updated Fig. 7 in Supplementary Information, as well as slight refinements in our decoherence estimates in the corresponding Section.

- What’s a “post-selected incoherent uniform random state”? Is it just post-selected counts from the maximally mixed state (i.e., from a uniform distribution)?

By way of explanation we place the following prescription for forming the post-selected incoherent uniform random state in the “Mutual information network analysis” section of the main text, to be highlighted in yellow:

“It is also useful to consider the effect of subjecting the incoherent uniformly random state to the same post-selection procedure as our experimental data. This can be done in three steps: i) form an incoherent uniformly random state with all 2^L amplitudes equal to $2^{-L/2}$, ii) set any basis state amplitudes to zero if the basis state has an eigenvalue under \mathcal{O} that differs from that of the QCA’s initial condition, and iii) renormalize the remaining amplitudes so that the state vector is properly normalized.”

- Is there a reason for using the word “emulate” rather than “simulate” when the latter is more common?

For semantic clarity when discussing our results, the word “simulate” does not readily or concisely distinguish between classical simulations and quantum hardware simulations. Therefore, in both the main text and supplementary information we use the term “emulate” to refer to executing our circuits on a classical computer. These emulations furnish a numerical benchmark against which to compare the same circuits executed on the quantum processor. The term is used to disambiguate the potentially dual meaning of “simulate”, which in this context refers to quantum simulation—simulation of one quantum mechanical system, here a QCA, by another one, here a gate-based quantum computer.

- The authors claim that the system’s dynamics occupy a fraction 1.08^L of the overall Hilbert space (due to the symmetry used for post-selection I believe). Is this just an upper-bound? Saying the state truly explores this region of the Hilbert space is a much stronger claim than simply saying it is constrained to this region.

We thank the reviewer for asking this question, as it has led us to refine our estimate of the scaling of the invariant-protected Hilbert space (what the reviewer refers to as the post-selection symmetry). As noted in a previous reply bullet, we find the bound on this protected Hilbert space now scales as $\sim 0.63 L^{1.91}$ (rather than the original $\sim 1.08^L$) due to a refined fitting procedure. To assess whether this bound is saturated or not, please find

the excerpt below, which we have placed in the “Post-Selection” section of Supplementary Information, highlighted in yellow, along with a new Fig. 9:

“Strictly-speaking, this scaling in the protected Hilbert space dimension is a bound on the system dynamics. In order to assess the extent to which the T_6 QCA’s dynamics actually saturate this bound, Fig. 9a shows the number of computational basis states occupied during the dynamics as a function of QCA cycle for the different system sizes (solid lines). QCA data is here generated by classical emulation. Dashed lines demarcate the $\dim(\mathcal{H}_O)$ scaling bounds. As can be seen over the first 30 QCA cycles, there always appears to be at least one point where the number of occupied basis states saturates the bound, with the two exceptions being $L = 21$ and $L = 23$. Fig. 9b shows the total fraction of the scaling bound occupied at *some* point along the first 30 cycles (colors correspond to those in panel a). Fig. 9b corroborates that the $\dim(\mathcal{H}_O)$ scaling bound is always saturated during at least one cycle in the first 30 cycles, with the exceptions again being $L = 21$ and $L = 23$, which reach $\approx 99\%$ and $\approx 98\%$ saturation, respectively.”

- Finally, a couple of suggestions for the figures: It would be helpful to put “t=0” rather than just 0 (maybe likewise for the other numbers) in Fig. 2b. It’s not clear at first glance what those numbers mean as-is. Also, it would be helpful to add labels to the color bars in Fig. 4d-g to spell out what quantity is being plotted.

We thank the reviewer for the suggestions. We have changed Fig. 2b and Fig. 4d-g in the main manuscript to reflect them.

Again, I think this is a nice paper, but I cannot recommend it for publication in this journal until some gaps are filled.

We thank the reviewer for their overall positive assessment of our manuscript. In the various replies above, we hope we have addressed the reviewer’s concerns to a sufficient extent as to permit publication in Nature Communications.

REVIEWERS' COMMENTS

Reviewer #2 (Remarks to the Author):

I would like to thank the authors for their detailed and broad answer. I agree with most of the scientific considerations in the message to reviewers (2 and 3) and I would stress that the originality of this experimental work was never questioned. Still, after careful reading of all authors points and resubmitted manuscript, I do not find scientific arguments showing that I missed some key aspects of this work in my previous report. My assessment about the importance/novelty/breakthrough is of course subjective but still stands.

Without entering in a point by point answer, but just as a brief comment, cluster states are also dynamically generated (in the corresponding experimental platforms) and are not just "theoretical" as the authors suggest in their answer. Also, I would suggest that some of the authors observations, for instance about the platform choice/relevance/interest, could be a relevant complement to further improve the manuscript.

Reviewer #3 (Remarks to the Author):

I am satisfied with the authors' response and the changes they made to the manuscript. I therefore recommend that this work be published in Nature Communications.

Reviewer Responses Round Two

Eric B. Jones et al.

June 2, 2022

Reviewer #2 (Remarks to the Author):

I would like to thank the authors for their detailed and broad answer. I agree with most of the scientific considerations in the message to reviewers (2 and 3) and I would stress that the originality of this experimental work was never questioned. Still, after careful reading of all authors points and resubmitted manuscript, I do not find scientific arguments showing that I missed some key aspects of this work in my previous report. My assessment about the importance/novelty/breakthrough is of course subjective but still stands.

We thank the reviewer for their positive assessment of the originality of our work. Other than by addressing the comments below, it is difficult to do more than we already have to allay the reviewer's concerns regarding their perceived lack of novelty of our manuscript. We note, however, that such concerns appear not to be wholly consistent with their positive appraisal of the work's originality. Nevertheless, we thank the reviewer for offering an extensive technical critique of the manuscript, which has strengthened it substantially.

Without entering in a point by point answer, but just as a brief comment, cluster states are also dynamically generated (in the corresponding experimental platforms) and are not just "theoretical" as the authors suggest in their answer.

We thank the reviewer for improving our understanding of this prior work. We have re-written the relevant paragraph in the main text, highlighted in yellow, which now reads:

"We note that complex-network analysis has already made a largely, though not wholly, theoretical impact on quantum information. One example arises in one-way quantum computing [Phys. Rev. A 68, 022312] in which complex, network-structured graph states [Nature Physics volume 15, pages 148–153 (2019)] are irreversibly transformed using projective measurements. Another occurs in the context of the quantum internet where communication channels between geographically-distant quantum devices are either imposed [Phys. Rev. Lett. 124, 210501] by fiber optic networks or implied [C Bonato et al 2009 New J. Phys. 11 045017] by satellite downlink capabilities. Notably, satellite-based quantum communication channels have recently been shown to support small-world connectivity [PRX Quantum 2, 010304]. Our work differs significantly from these examples. Where one-way quantum computing has experimentally realized complex graph states by design via projective measurement, our work shows that QCA dynamically generate them in an emergent fashion. Where the quantum internet considers geographic networks, our networks of correlations emerge from unitary dynamics without any notion of physical distance except the locality of interactions. Finally, the complex networks occurring herein emerge on a generally-programmable, gate-model quantum processor, that is, an experimental platform with real-world constraints, such as processor noise, not necessarily present in prior theoretical work and different in nature than in prior experimental work."

Also, I would suggest that some of the authors observations, for instance about the platform choice/relevance/interest, could be a relevant complement to further improve the manuscript.

We agree with the reviewer’s suggestion and place the following (taken from the response to the reviewer’s associated comment in the first round of reviews) in the Supplementary Information under the heading “Comment on Simulation Platform Choice” (Note that this change is not highlighted in yellow as previously, since the Supplementary Information file must be finalized at this stage of review):

“Here, we comment briefly on our choice of simulation platform for the experimental realization of QCA. First, we justify the use of a universal, gate-model processor against an analog quantum simulator. Chains of Rydberg atoms up to 51 and trapped ions up to 53 sites long have been used simulate the many-body dynamics of kinetically-constrained (sometimes called PXP [C. J. Turner, 2018, doi:<https://link.aps.org/doi/10.1103/PhysRevB.98.155134>]) and quantum Ising-type spin models out to microseconds and milliseconds long, respectively [H. Bernien et al., 2017, doi:<https://doi.org/10.1038/nature24622>] [J. Zhang, 2017, doi:<https://doi.org/10.1038/nature24654>]. These are the existing experiments that most closely approximate ours on the analog simulation side of the field. While these experiments exhibit larger qubit counts and, in some instances, better coherence properties, analog quantum simulators are highly-constrained in the systems they can simulate by the native degrees of freedom and interactions of the underlying simulator. That is, while particular analog simulation platforms may be appropriate for simulating particular quantum cellular automata, it would be very difficult to design one as a general-purpose QCA simulator. This reflects the need for universal, gate-model machines writ large. As such, though it is required in our experiment to combat higher noise levels than those found in analog simulators through a variety of techniques, some of which thereby limiting the observables we are able to calculate, our approach to QCA simulation is generalizable not just to other QCA in one-dimension, but to those of higher dimension and greater connectivity as well, given sufficiently improved noise characteristics of the underlying quantum processor.

Second, having addressed the issue of analog versus digital quantum simulation, we remark upon our particular choice of digital platform. There currently exist various approaches to constructing gate-model quantum computers, among them superconducting, trapped ion, neutral atom, photonic, and topological qubit-based processors. Each of these approaches involves design trade-offs. At the time of performing our experiment, the large majority of cloud-accessible gate-model quantum computers were either trapped ion (e.g., Quantinuum, IonQ) or superconducting machines (e.g., Google, Rigetti, IBM). The capability of a given quantum processor involves trade-offs in terms of qubit count, qubit connectivity, decoherence times (relative to gate execution times), native gate expressiveness, and gate and state-preparation and measurement error rates. While trapped ion two-qubit error rates can be somewhat lower than superconducting error rates— e.g., $\sim 0.3\%$ for Quantinuum’s H1.1 processor— and are fully-connected due to the ability to shuttle ions, cloud accessible trapped ion processors remain rather size-limited with respect to superconducting processors [T. Lubinski et al., 2021, doi:<https://doi.org/10.48550/arXiv.2110.03137>]. For example, at the time our experiment was performed, Quantinuum’s cloud-accessible processor could load up to 12 ions and IonQ’s processor could load up to 11 ions. By comparison, we have simulated QCA as large as 23 qubits long. Moreover, we did not require all-to-all connectivity for our QCA circuits since all interactions were nearest-neighbor. We also note that our employment of the Floquet calibration technique allowed for the additional suppression of two-qubit gate error in our system beyond the $\sim 1.4\%$ rate determined by cross-entropy benchmarking. As for superconducting processors, the Rainbow and Weber processors on which we ran our experiments are at least as performant

as those offered by other superconducting processor vendors such as IBM or Rigetti. In summary, it is unlikely that running our QCA circuits on a different current-generation, gate-model quantum processor would have resulted in a positive step-change in terms of fidelity, although we would be fascinated to see our results repeated or improved upon on a different architecture.”

Reviewer #3 (Remarks to the Author):

I am satisfied with the authors’ response and the changes they made to the manuscript. I therefore recommend that this work be published in Nature Communications.

We thank the reviewer for their recommendation to publish, and for their previous comments, which significantly improved the manuscript.